# BEYOND SHORT STEPS IN FRANK-WOLFE ALGORITHMS

**David Martínez-Rubio**
IMDEA Software Institute,
Madrid, Spain
david.martinezrubio@imdea.org

**Sebastian Pokutta**
Zuse Institute Berlin and
Technische Universität Berlin, Berlin, Germany
pokutta@zib.de

## ABSTRACT

We introduce novel techniques to enhance Frank-Wolfe algorithms by leveraging function smoothness beyond traditional short steps. Our study focuses on Frank-Wolfe algorithms with step sizes that incorporate primal-dual guarantees, offering practical stopping criteria. We present a new Frank-Wolfe algorithm utilizing an optimistic framework and provide a primal-dual convergence proof. Additionally, we propose a generalized short-step strategy aimed at optimizing a computable primal-dual gap. Interestingly, this new generalized short-step strategy is also applicable to gradient descent algorithms beyond Frank-Wolfe methods. Empirical results demonstrate that our optimistic algorithm outperforms existing methods, highlighting its practical advantages.

## 1 INTRODUCTION

We are interested in solving the following optimization problem:

$$\min_{x \in \mathcal{X}} f(x),$$

where $\mathcal{X}$ is a compact convex set and $f$ is a convex and $L$-smooth function. The Frank-Wolfe (FW) algorithm (Frank & Wolfe, 1956), also known as the conditional gradient algorithm (Levitin & Polyak, 1966), is a key algorithm for this problem class, particularly for problems where projection onto a constraint set is computationally expensive, as it does not require projections. It leverages a linear minimization oracle (LMO) for $\mathcal{X}$, which upon presentation with a linear function $c$ returns $v \leftarrow \arg\min_{v \in \mathcal{X}} \langle c, v \rangle$, and a gradient oracle for $f$, which given a point $x \in \mathrm{dom}(f)$ returns $\nabla f(x)$. Given its low cost per iteration it is often highly effective in various machine learning applications. These include optimal transport (Luise et al., 2019), neural network pruning (Ye et al., 2020), adversarial attacks (Chen et al., 2020), non-negative matrix factorization (Nguyen et al., 2022), particle filtering (Lacoste-Julien et al., 2015), and distributed learning (Bellet et al., 2015), among others.

A more general version of the Frank-Wolfe algorithm can handle the optimization of $f(x) + \psi(x)$ for a convex function $\psi$. In this case however, we require access to a gradient oracle for $f$ and an oracle that can solve $\arg\min_v \{\langle w, v \rangle + \psi(v)\}$ for any $w \in \mathbb{R}^d$, as discussed in (Nesterov, 2018). We recover the classical FW algorithm by choosing $\psi(x)$ as the indicator function of the set $\mathcal{X}$.

In this work, we will focus on primal-dual analyses for FW algorithms. In contrast to classical analyses where a step-size strategy and corresponding convergence rate need to be heuristically estimated and then proven by induction, in the primal-dual setup they emerge as a natural consequence of the analysis. This approach does not only provide tighter dual gaps and hence stopping criteria but also enhances the convergence properties of the algorithm. In particular, rather than considering primal progress without clear indication how good the solution is, primal-dual progress refers to the reduction in the primal-dual gap, which is a measure of how close the current solution is to optimality; this will be our measure here. A well-known strategy for FW algorithms is the so-called short step, which essentially chooses the step size to maximize primal guaranteed progress from the descent lemma, see (Frank & Wolfe, 1956)); and also (Braun et al., 2022) for an in-depth discussion.

We introduce the concept of primal-dual short steps, that maximize primal-dual guaranteed progress based on a model obtained from the primal-dual analysis. We also show that this new step-size

strategy is also applicable to gradient descent algorithms. Moreover, recent advancements have introduced adaptive step-size strategies, such as, e.g., (Pedregosa et al., 2020) (see also (Pokutta, 2024) for a numerically improved version), which further refine the short-step. These strategies aim to decrease a model of the function adaptively, enhancing the algorithm's performance and stability and in particular allow for leveraging local curvature information. However, they come at the extra cost of a line search like procedure and are subject to the same lower bounds as the classical short steps (Guélat & Marcotte, 1986).

## MAIN CONTRIBUTIONS

**Optimistic Frank-Wolfe Algorithm**   We introduce a novel Frank-Wolfe method that leverages the concept of optimism (Rakhlin & Sridharan, 2013; Steinhardt & Liang, 2014). This algorithm is designed to adapt effectively to varying conditions and provides a robust analysis of the convergence rate associated with a primal-dual gap.

**Primal-Dual Short Steps**   We propose a new class of step-size rules for existing Frank-Wolfe algorithms, termed primal-dual short steps. These steps are based on the sequential minimization of a primal-dual gap defined in the algorithm's analysis. This approach is flexible and extends to gradient descent algorithms, allowing for line search over the primal-dual gap in both algorithm classes.

**Numerical Experiments**   We conduct numerical experiments that showcase the practical advantages of the proposed optimistic Frank-Wolfe algorithm. In our tests, the results consistently show the effectiveness of this new variant in minimizing the primal-dual gap more efficiently than traditional methods. In particular, in most experiments the empirical order of convergence is better, as demonstrated by the steeper slope in a log plot.

Additionally, we have obtained primal-dual analyses for algorithms and results where so far only classical primal convergence analysis where available, often improving constants and providing effective stopping criteria for these algorithms. These findings are detailed in Appendix E.

## RELATED WORK

The Frank-Wolfe algorithm, introduced by Frank & Wolfe (1956) where also short steps were introduced, is a foundational method in projection-free optimization, particularly for problems involving convex constraints. Levitin & Polyak (1966) further extended the method, which they referred to as *conditional gradient algorithm*. A significant advancement for the Frank-Wolfe algorithm was made by Jaggi (2013), who introduced the FW gap and provided convergence guarantees on this measure, marking the first instance of FW guarantees for a primal-dual gap.

The development of primal-dual methods for conditional gradient algorithms has been explored in works such as Nesterov (2018) and Diakonikolas & Orecchia (2019), which have contributed to the broader understanding of these algorithms. Additionally, Abernethy & Wang (2017) described a Frank-Wolfe variant that incorporates cumulative gradients, albeit with equal weights, resulting in an additional logarithmic factor in complexity.

Several methods apply the LMO at a convex combination of previously obtained directions. In particular, variants of the Frank-Wolfe algorithm have been developed by Mokhtari et al. (2020), Lu & Freund (2021), and Négiar et al. (2020) using stochastic gradients. Notably, the analysis of Lu & Freund (2021) reduces to the optimal rate in the deterministic case, up to constant factors. Wang et al. (2024) also presented a cumulative-gradient FW algorithm, with additional logarithmic factors. Further contributions include the work of Li et al. (2021), who provided a primal-dual analysis of the deterministic algorithm, heavy-ball Frank-Wolfe (HB-FW), using a convex combination of gradients.

In the context of online learning, the concept of optimism, which involves using hints such as past losses to predict future losses, originated in (Azoury & Warmuth, 2001; Chiang et al., 2012; Rakhlin & Sridharan, 2013). This idea had already been developed in saddle point optimization, as seen in the work of Popov (1980). A duality was found and further explored between mirror descent and Frank-Wolfe algorithms by Bach (2015) and Peña (2019). Wang et al. (2024) and Gutman & Peña (2023) have also shown that FW and mirror descent, among other methods, can be unified under a

common framework. In this work, we generalize an anytime online-to-batch conversion (Nesterov & Shikhman, 2015; Cutkosky, 2019) for connecting FW algorithms with online learning ones.

Works that have studied modifications of the gradients to obtain faster convergence of FW algorithms and linking them back to gradient descent methods and gradient mappings include Combettes & Pokutta (2020); Mortagy et al. (2020) and in Diakonikolas et al. (2020) local acceleration in the Nesterov sense has been demonstrated for FW algorithms for strongly convex functions while whether local acceleration is possible for non-strongly convex but smooth functions remains an open question.

## 2    PRELIMINARIES AND GROUNDWORK

In this section, we introduce the necessary notation and an overview of some FW algorithms. In the following let $\mathcal{X}$ be a compact convex set with diameter $D$. Further, let $f : \mathcal{X} \to \mathbb{R}$ be a differentiable function defined on an open set containing $\mathcal{X}$. We assume that $f$ is convex and $L$-smooth on $\mathcal{X}$ with respect to a norm $\|\cdot\|$. This means that for all $x, y \in \mathcal{X}$, the following inequality holds:

$$0 \leq f(y) - f(x) - \langle \nabla f(x), y - x \rangle \leq \frac{L}{2}\|x - y\|^2.$$

These two conditions imply $f$ has an $L$-Lipschitz gradient $\|\nabla f(x) - \nabla f(y)\|_* \leq L\|x - y\|$, where $\|\cdot\|_*$ denotes the dual norm.

If not stated otherwise let $x^* \in \arg\min_{x \in \mathcal{X}} f(x)$ be a minimizer of $f$ over $\mathcal{X}$. Further, let $I_{\mathcal{X}}(x)$ be the indicator function of the set $\mathcal{X}$ that is $0$ if $x \in \mathcal{X}$ and $+\infty$ otherwise, and let $1_B$ be the event indicator function that is $1$ if the event $B$ holds and $0$ otherwise. We denote $[T] \stackrel{\text{def}}{=} \{1, 2, \ldots, T\}$. We denote by $\partial \psi(x)$ the subdifferential of $\psi$ at $x$ so that if $g \in \partial \psi(x)$, we have $\psi(y) \geq \psi(x) + \langle g, y - x \rangle$.

We provide now a sketch of the proof structure for some primal-dual analysis of FW as well as HB-FW and other methods, and in Appendix B we provide full details of this overview. With respect to (Li et al., 2021), we provide a slightly more general analysis of HB-FW by allowing for a composite term and we provide an analysis of FW when decreasing regularization is used. We start by defining a lower bound $L_t$ on the optimal value that we have access to at time $t$, and for iterates $x_t$, $t \geq 1$, we define a primal-dual gap as

$$G_t \stackrel{\text{def}}{=} f(x_{t+1}) - L_t \tag{1}$$

where $f(x_{t+1})$ is one step ahead of the lower bound, which is a subtle shift that differs slightly from known analysis and allows us to get very simple primal-dual analyses of and convergence results for the new and old algorithms we present in this work. In the sequel, if not stated otherwise, let $a_t > 0$ to be determined later and define $A_t = A_{t-1} + a_t = \sum_{\ell=0}^{t} a_\ell$. For instance, for FW and HB-FW we use the following lower bounds, or rather, the choice of lower bound defines the iterate $v_t$ of the algorithm:

$$A_t f(x^*) \overset{①}{\geq} \sum_{\ell=0}^{t} a_\ell f(x_\ell) + \sum_{\ell=0}^{t} a_\ell \langle \nabla f(x_\ell), x^* - x_\ell \rangle$$

$$\overset{②}{\geq} \sum_{\ell=0}^{t} a_\ell f(x_\ell) + \min_{v \in \mathcal{X}} \left\{ \sum_{\ell=0}^{t} a_\ell \langle \nabla f(x_\ell), v - x_\ell \rangle \right\} \overset{\text{def}}{=} A_t L_t^{\text{HB}} \tag{2}$$

$$\overset{③}{\geq} \sum_{\ell=0}^{t} a_\ell f(x_\ell) + \sum_{\ell=0}^{t} a_\ell \min_{v \in \mathcal{X}} \langle \nabla f(x_\ell), v - x_\ell \rangle \overset{\text{def}}{=} A_t L_t^{\text{FW}} = \sum_{\ell=0}^{t} a_\ell f(x_\ell) - \sum_{\ell=0}^{t} a_\ell g(x_\ell).$$

where the term above $g(x_\ell) \stackrel{\text{def}}{=} \max_{v \in \mathcal{X}} \langle \nabla f(x_\ell), x_\ell - v \rangle \ (\geq f(x_\ell) - f(x^*))$ is the so-called FW gap, which is also a primal dual gap, the best of which for $\ell \in [t]$ was shown to converge by Jaggi (2013) at a $O(\frac{LD^2}{t})$ rate for FW. Above ① holds by convexity, ② takes a min. This expression is $A_t$ times the lower bound $L_t^{\text{HB}}$ in HB-FW is and its $\arg\min$ corresponds to the iterate $v_t$ of HB-FW, cf. Algorithm 1, whereas after ③ we have $A_t$ times the lower bound $L_t^{\text{FW}}$ used in the FW algorithm and the $\arg\min$ of the $\ell$-th summand is the iterate $v_t$ of FW, cf. Algorithm 1.

---

**Algorithm 1** Frank-Wolfe and Heavy-Ball FW algorithms

---

**Input:** Function $f$, feasible set $\mathcal{X}$, initial point $x_0$. Weights $a_t, \gamma_t$.

1: **for** $t = 0$ to $T$ **do**
2:     $\diamond$ Choose either 3 (HB-FW) or 4 (FW) for the entire run:
3:     $v_t \leftarrow \underset{v \in \mathcal{X}}{\arg\min} \sum_{i=1}^{t-1} a_i \langle \nabla f(x_i), v \rangle$
4:     $v_t \leftarrow \underset{v \in \mathcal{X}}{\arg\min} \langle \nabla f(x_t), v \rangle$
5:     $x_{t+1} \leftarrow (1 - \gamma_t) x_t + \gamma_t v_t$
6: **end for**

---

If we now show a bound

$$A_t G_t - A_{t-1} G_{t-1} \leq E_t \text{ for all } t \geq 0, \tag{3}$$

then we have $f(x_{t+1}) - f(x^*) \leq G_t \leq \frac{1}{A_t} \sum_{i=0}^{t} E_i$. Our aim is thus to have small $E_t$ and large $A_t$. Note $A_{-1} = 0$ by definition. If $f$ is $L$-smooth with respect to $\|\cdot\|$ and $D \overset{\text{def}}{=} \max_{x,y \in \mathcal{X}} \|x - y\|$, then choosing $a_t = 2t + 2$, $\gamma_t = a_t/A_t$ one can show that both FW and HB-FW in Algorithm 1 satisfy (3) with $E_t = \frac{LD^2 a_t^2}{2A_t}$ and consequently $G_t \leq \frac{2LD^2}{t+2}$, cf. Appendix B. The term $f(x_{t+1}) - f(x^*)$ cannot be computed in general since we usually do not have access to the value $f(x^*)$. However, the primal-dual bound $G_t$ is computable and thus it can be used as a stopping criterion.

**Remark 2.1** (Alternative step-size strategies). *The analysis uses the guaranteed primal progress, that is, the descent that is guaranteed on $f(x_{t+1}) - f(x_t)$, in particular with the choice $\gamma_t = \frac{a_t}{A_t} = \frac{2}{t+2}$ by using the upper bound that the smoothness inequality provides. Any step size strategy whose primal progress is greater than this guaranteed progress yields a better convergence rate. For instance, the so-called short step-size strategy, maximizes this guaranteed progress along the segment joining $x_t$, and $v_t$. Indeed the best lower bound on $f(x_t) - f(x_{t+1})$ given by smoothness is*

$$\max_{\gamma \in [0,1]} \left\{ \gamma \langle f(x_t), x_t - v_t \rangle + \gamma^2 \frac{L \|x_t - v_t\|^2}{2} \right\}.$$

*And the $\arg\max$ is $\gamma = \min\{1, \frac{\langle \nabla f(x_t), x_t - v_t \rangle}{L \|x_t - v_t\|^2}\}$, which is the short step-size. Another alternative is performing a line search over the segment joining $x_t$ and $v_t$. Even if the line search is performed with an error at iteration $t$, this error only contributes additively to the corresponding $E_t$, and so its global impact can be easily quantified.*

Although short steps induce monotonic primal progress by definition, they do not necessarily induce monotonic primal-dual progress, which is the important measure to look at when we require a stopping criterion. To remedy this, in Section 4 we introduce a new primal-dual short step strategy that induces monotonic primal-dual progress.

On the other hand all of previous FW approaches exploit smoothness of $f$ in the same way: a quadratic upper bound on the function is computed and the yielded guaranteed progress is used to compensate from some other errors in the analysis. In Section 3, we introduce a framework that goes beyond this by making use of optimism. The algorithm has the potential of better adapt to the environment, as we show in the experiments section, where Optimisitic Frank-Wolfe performs better than other approaches.

## 3   AN OPTIMISTIC FRANK-WOLFE ALGORITHM

In this section, we propose a new Frank-Wolfe method that at each iteration, uses a prediction of the next gradient in order to minimize a suitable regularized lower model of the objective. The more accurate this prediction is, the better the resulting convergence rate. For functions with Lipschitz continuous gradients, the previous gradient serves as a sufficiently accurate predictor for the next one. We sketch the main ideas of our new algorithm and provide all details in Appendix C.

The algorithm and analysis is based on optimistic versions of the Online Mirror Descent (OMD) algorithm and of the Follow the Regularized Leader (FTRL) algorithm. We provide a slight generalization over these online learning algorithms in order to allow for non-differentiable regularizers, in order to cover typical cases where Frank-Wolfe is applied. To that effect, we make use of the following

Bregman divergence definition, that specifies a subgradient of $\phi$ of the regularizer for its definition: $D_\psi(x, y, \phi) \stackrel{\text{def}}{=} \psi(x) - \psi(y) - \langle \phi, x - y \rangle$, where $\phi \in \partial\psi(y)$. Note that in the pseudocode of Algorithm 2 we write $D_\psi(v, v_{t-1}, \phi_t \in \partial\psi(v_{t-1}))$ to mean that the algorithm can use $D_\psi(v, v_{t-1}, \phi_t)$ for any $\phi_t \in \partial\psi(v_{t-1})$.

Online learning algorithms typically use regularizers $\psi$ that are strongly convex or enjoy any other curvature property such as uniform convexity, in order to obtain a low enough regret. Even though we do not assume strong convexity or any other curvature property of the regularizer $\psi$ (in fact, $\psi$ can be 0 in the feasible set), we show that we can apply an optimistic approach that leads to the optimal convergence rate in this setting, and we show in Section 5 that our algorithm *empirically outperforms other approaches*.

The starting point of the method, as in (2), is defining a lower bound on the optimal value, which we do by taking inspiration on the anytime online-to-batch conversion of Cutkosky (2019), that connects the regret of online learning algorithms with convergence guarantees of optimization methods. This naturally leads to the definition $x_t = \frac{1}{A_t}\sum_{i=1}^{t} a_i v_i = \frac{a_t}{A_t}v_t + \frac{A_{t-1}}{A_t}x_{t-1}$ for all $t \geq 1$ in Algorithm 2. We are also able to show that at each iteration, the theory allows for making use of a point $y_{t-1}$ with lower function value than the previous $x_{t-1}$, in place of using $x_{t-1}$, and we show that this does not hurt the convergence guarantee, so it allows for heuristics like performing line search over the segment $x_{t-1}$ and $v_t$, or other future heuristics that may be built on top of this algorithm. The definition of $v_t$ comes from the optimistic online learning algorithmic schemes.

In a similar fashion to the one in (1), using said lower bound we define a primal-dual gap that we denote $G_t^{\text{OP}}$ for both variants of the algorithm, cf. Appendix C.2. We provide guarantees on this primal-dual gap by the regret of the optimistic procedure, gradient Lipschitzness of the function, along with the loss weighting and compactness of the domain.

We use an optimistic FTRL algorithm (Algorithm 2) or an optimistic MD algorithm (Algorithm 2) with *constant step size*, designed to work for subdifferentiable losses. Interestingly, while it is well-known that for constant step size and unconstrained problems, FTRL and OMD have the same updates, we note the more general property that in the constrained setting, FTRL is an instance of OMD with subdifferentiable regularizers for a precise choice of subgradients, cf. Remark C.5.

---

**Algorithm 2** Optimistic Frank-Wolfe algorithms

---

**Input:** Convex subdifferentiable regularizer $\psi$ such that $\arg\min_x\{\langle w, x \rangle + \psi(x)\}$ exists for all $w \in \mathbb{R}^d$. A convex function $f$, differentiable and $L$-smooth with respect to a norm $\|\cdot\|$ in $\text{dom}(\psi)$. Initial point $x_0 = v_0 \in \text{dom}(\psi)$. Unbiased stochastic gradient oracle $\mathfrak{O}(x)$ with finite variance $\sigma^2$.

1: $A_0 \leftarrow 0; a_0 \leftarrow 0; g_0 = 0; \quad g_1 = \mathfrak{O}(x_0)$
2: **for** $t \leftarrow 1$ **to** $T$ **do**
3: $\quad a_t \leftarrow 2t; A_t \leftarrow A_{t-1} + a_t = \sum_{i=1}^{t} a_i = t(t+1)$
4: $\quad \diamond$ Choose either 5 or 6 for the entire run, for $\phi_t \in \partial\psi(v_{t-1})$:
5: $\quad v_t \leftarrow$ point in $\arg\min_{v \in \mathbb{R}^d}\{\sum_{i=1}^{t-1} a_i\langle g_{i+1}, v \rangle + a_t\langle g_t, v \rangle + \psi(v)\}$
6: $\quad v_t \leftarrow$ point in $\arg\min_{v \in \mathbb{R}^d}\{a_{t-1}\langle g_t - g_{t-1}, v \rangle + a_t\langle g_t, v \rangle + D_\psi(v, v_{t-1}; \phi_t)\}$
7: $\quad y_{t-1} \leftarrow x_{t-1}$ $\quad\quad\quad \diamond$ Also valid: $y_{t-1} \leftarrow$ point such that $f(y_{t-1}) \leq f(x_{t-1})$ (in det. algs.)
8: $\quad x_t \leftarrow \frac{A_{t-1}}{A_t}y_{t-1} + \frac{a_t}{A_t}v_t$ $\quad\quad\quad\quad\quad\quad \diamond$ that is, $x_t \leftarrow \frac{t-1}{t+1}y_{t-1} + \frac{2}{t+1}v_t$
9: $\quad g_{t+1} \leftarrow \mathfrak{O}(x_t)$
10: **end for**
11: **return** $x_T$.

---

One subtlety of $G_t^{\text{OP}}$ is that it is not directly computable as is, since the direction we apply the LMO depends on the hint that we choose, that is, a prediction for the next gradient. We can compute a simple close bound of it at every iteration, or if we want to compute it after certain number of iterations, we can do it by performing an extra LMO, cf. Remark C.6. The guarantee we obtain on the algorithm is the following. We allow for a stochastic oracle that computes an unbiased estimate of the gradient with the finite variance assumptions $\mathbb{E}[\|\mathfrak{O}(x) - \nabla f(x)\|^2] < \sigma^2$. The deterministic result corresponds to $\sigma = 0$.

**Theorem 3.1.** [↓] *Let $\mathcal{X}$ be compact and convex, and let $\psi : \mathcal{X} \to \mathbb{R}$ be a closed convex function, subdifferentiable in $\mathcal{X}$. Let $f$ be convex and $L$-smooth in the set $\mathcal{X}$ of diameter $D$ with respect to*

*a norm $\|\cdot\|$ that we access via an unbiased stochastic gradient oracle $\mathfrak{O}(x)$ with finite variance $\sigma^2 \geq 0$. The iterates $x_t$ of Algorithm 2 satisfy:*

$$\mathbb{E}[f(x_t) - f(x^*)] \leq \mathbb{E}[G_t^{\mathrm{OP}}] \leq \frac{\psi(x^*) - \psi(x_1)}{t(t+1)} + \frac{4LD^2}{t+1} + \frac{\sqrt{2}\sigma D}{2},$$

*for the variant in Line 5. For the variant in Line 6 we obtain the same except that $\psi(x^*) - \psi(x_1)$ is substituted by $D_\psi(x^*, x_0; \phi_0)$, where $\phi_0 \in \partial\psi(x_0)$.*

We note that the term involving $\psi$ is just a consequence of our choice of step sizes, which we used for simplicity. It is possible to keep the $O(\frac{LD^2}{t+1})$ rate and have an arbitrarily fast polinomial-on-$t$ decay on the term involving $\psi$ by making a different choice of step sizes, see Appendix C.2.

## 4 PRIMAL-DUAL SHORT STEPS

A key step of the analysis in most Frank-Wolfe algorithms for smooth problems, such as in Section 2, consists of using the smoothness inequality in order to guarantee some descent that compensates other per-iteration errors that appear. Several step-size rules have been devised, that may change depending on the specific setting. However, three families of step sizes stand out in almost every setting, which correspond to the ones we discussed in Section 2: (A) open-loop step-sizes, that only depend on the iteration count, (B) short steps, that minimize the upper bound given by the last computed gradient $\nabla f(x_t)$ and the smoothness inequality with respect to some norm, along the segment in between the current point $x_t$ and the computed Frank-Wolfe vertex $v_t \in \arg\max_{v \in \mathcal{X}}\{\langle \nabla f(x_t), v \rangle\}$, and (C) line search in the aforementioned segment to maximize primal progress.

In the sequel, we devise a new class of step sizes, which are a generalization of (B). The idea of (B) is to greedily maximize the *guaranteed* primal progress along the segment in between $x_t$ and $v_t$, where we know we are feasible. The key idea of our new step-size rule consists of taking the primal-dual gap (1) that is defined for the analyses with the structure of (3), and choosing the step size in order to maximize the *guaranteed* progress in terms of this primal-dual gap.

We also show that our primal-dual gap bound at iteration $t$ is convex with respect to the step size, which implies that one can efficiently do a line search to maximize the primal-dual progress. In order to show the flexibility of this paradigm, we also generalize these ideas to the gradient descent algorithm.

### 4.1 PRIMAL-DUAL SHORT STEPS FOR FW ALGORITHMS

Let us consider first the case of FW and HB-FW. In this section, we denote $G_t$, and $x_t$ and $v_t$ the respective primal-dual gaps, and iterates. From the analyses in Appendix B, cf. (8) or (11), we have the following

$$A_t G_t - A_{t-1} G_{t-1} \leq \frac{La_t^2}{2A_t}\|v_t - x_t\|^2,$$

and defining $\gamma_t \stackrel{\text{def}}{=} a_t/A_t$, dividing the above by $A_t$ on both sides, and rearranging gives

$$G_t \leq (1 - \gamma_t)G_{t-1} + \gamma_t^2 \frac{L}{2}\|v_t - x_t\|^2. \tag{4}$$

The right-hand side is minimized for the choice

$$\gamma_t = \min\left\{1, \frac{G_{t-1}}{L\|v_t - x_t\|^2}\right\}, \tag{5}$$

which is what we refer to as the *primal-dual short step* for these two algorithms, and focuses on maximizing guaranteed progress of the primal-dual gap, as discussed above. We show that this step-size is sound in the sense that we still keep the optimal convergence guarantees of other approaches. We also show that we can perform a line search over the primal-dual gap bound obtained before using the smoothness inequality in the analyses.

**Proposition 4.1.** [↓] *Let $f$ be convex and differentiable. The FW and HB-FW algorithms satisfy*

$$G_t \leq (1 - \gamma)G_{t-1} - \gamma\langle\nabla f(x_t), v_t - x_t\rangle - f(x_t) + f((1 - \gamma)x_t + \gamma v_t), \tag{6}$$

*for all $\gamma \in [0, 1], t > 1$ and the RHS is convex on $\gamma$. If further $f$ is L-smooth w.r.t. a norm $\|\cdot\|$, and $D \stackrel{\text{def}}{=} \max_{x,y\in\mathcal{X}} \|x - y\|$, then using (5) or line search on the RHS of (6), we obtain: $G_t \leq \frac{4LD^2}{t+2}$.*

The convergence above for the line search is derived from the one for the primal-dual short step, since the right hand side of (5) upper bounds the one of (6).

We note that, naturally, if we define the gap as $G_t \stackrel{\text{def}}{=} f(x_{t+1}) - f(x^*)$, which is the primal gap, then analyzing the algorithm with the strategy in (3), yields that the primal-dual short steps become regular short steps. In that case, even though we do not know the value $f(x^*)$ in the gap definition, the step can still be defined without this knowledge, cf. Remark D.1.

## 4.2 PRIMAL-DUAL SHORT STEPS FOR GRADIENT DESCENT

We now extend our primal-dual short steps to gradient descent (GD) whose updates are given by $x_{t+1} \leftarrow x_t - a_t\nabla f(x_t)$. We use the Euclidean norm in this section for the problem $\min_{x\in\mathbb{R}^d} f(x)$, where $f$ is convex and $L$-smooth. One possible analysis of GD relies on defining a primal-dual gap similarly to the one in Appendix B.2, and then show $A_tG_t \leq A_{t-1}G_{t-1}$ for $a_t = 1/L$ and $A_1G_1 = \frac{1}{2}\|x_1 - x^*\|_2^2$, for an initial point $x_1$ and a minimizer $x^*$. We provide a sketch of the steps that we take and leave the details to Appendix D.1. The lower bound on $f(x^*)$ that we use is

$$A_t f(x^*) \geq \sum_{i=1}^t a_i f(x_i) + \sum_{i=1}^t a_i\langle\nabla f(x_i), x_{t+1} - x_i\rangle + \frac{1}{2}\|x_{t+1} - x_1\|_2^2 - D^2 \stackrel{\text{def}}{=} A_t L_t,$$

where $D$ is an upper bound on the initial distance to a minimizer $\|x^* - x_1\|_2$. Defining the gap as $G_t = f(x_{t+1}) - L_t$, we arrive to

$$G_t \leq \frac{A_{t-1}}{A_t}G_{t-1} + \|\nabla f(x_t)\|_2^2 \left(-\frac{a_t A_{t-1}}{A_t} + \frac{a_t^2 L}{2} - \frac{a_t^2}{2A_t}\right),$$

for $t > 1$ and a similar expression for $t = 1$. Recall that $A_t \stackrel{\text{def}}{=} \sum_{i=1}^t a_t$. So at this stage, one can optimize $a_t$ in order to minimize the right hand side, by solving a simple cubic equation, which is what we term the primal-dual short steps for GD. We show that the optimal value satisfies $a_t \geq \frac{1}{2L}$, which ultimately yields to a convergence rate no slower than $G_t \leq \frac{LD^2}{t}$, as we formalize in the following. We also show, as above, that we can perform a line search for minimizing a better bound on the primal-dual gap, yielding no worse convergence rates. Recall that $f(x_{t+1}) - f(x^*) \leq G_t$.

**Proposition 4.2.** [↓] *Let $f$ be convex and differentiable. The primal-dual gap of GD is bounded by (25) which is convex on the step $a_t$. If further $f$ is L-smooth with respect to $\|\cdot\|_2$, GD using the primal-dual short step-size or line search on the aforementioned bound then the step size satisfies $a_t \geq \frac{1}{2L}$ and we have $G_t \leq \frac{LD^2}{t}$.*

A notable difference between the line search on these primal-dual bounds in GD and FW algorithms is that convexity was shown in Proposition 4.1 to hold with respect to the parameter $\gamma$ which corresponds to $\frac{a_t}{A_t}$ whereas for GD in Proposition 4.2, the convexity is shown to hold for the parameter $a_t$.

## 5 EXPERIMENTS

Here we provide experiments demonstrating the good performance of the optimistic variant. All experiments were performed in Julia based on the `FrankWolfe.jl` package run on a MacBook Pro with an Apple M1 chip with Julia 1.11.1. The code will be made publicly available upon publication. In our experiments we compare the optimistic FW variant (with OFTRL) with the heavy ball variant, and the vanilla FW algorithm. For the FW variant, apart from the agnostic step size rule $\gamma_t \stackrel{\text{def}}{=} \frac{2}{t+2}$ we also consider the adaptive line search (indicated by `adaptive` in the plots) from (Pokutta, 2024), a numerically improved variant of (Pedregosa et al., 2020), which is the default line search in the `FrankWolfe.jl` package. As shown in (Guélat & Marcotte, 1986), line search is subject to a

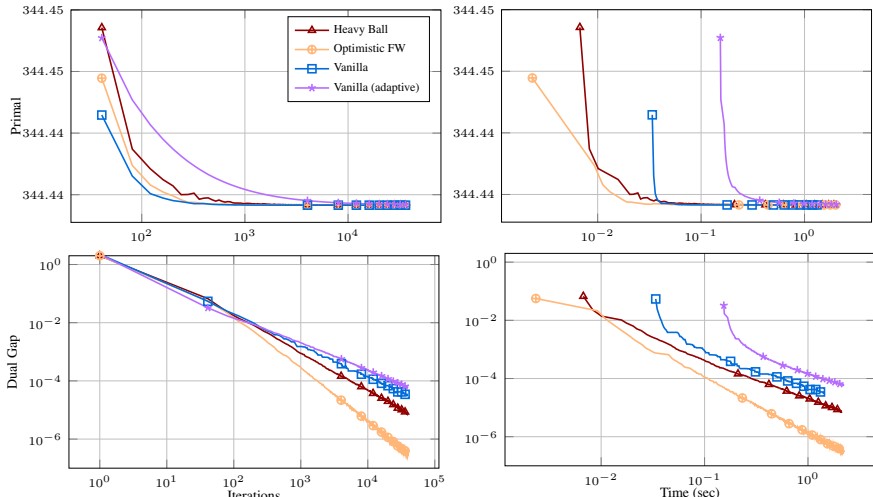

Figure 1: Comparison over the probability simplex with objective $f(x) = \|x - x_0\|_2^2$, where $x_0$ is a random point outside the probability simplex. The optimistic method converges faster in iterations and time.

lower bound that does not hold for, e.g., open-loop step-sizes and in particular the heavy ball variant and the optimistic variants. In fact, there are cases where the line search variant is slower than the vanilla variant as asymptotically shown in (Bach, 2021) and later in (Wirth et al., 2023; 2024) for the non-asymptotic case; we observe a similar behavior in our experiments as they satisfy the conditions of (Wirth et al., 2023; 2024); see also (Kerdreux et al., 2021).

Our numerical experiments are over various polytopes $P$, so that in particular $\psi(x) \stackrel{\text{def}}{=} I_P(x)$ is the indicator function of the respective polytope. In the following, we will use the dual gap the definition of $G_t \stackrel{\text{def}}{=} f(x_t) - L_t^{FW}$, as in (2). In the case of the heavy ball variant from Appendix B.1 we set $v_i = v_t$, for all $i \le t$, and $v_t \in \arg\min_{v \in P} \sum_{i=0}^{t} \frac{a_i}{A_t} f(x_i) + \sum_{i=0}^{t} \frac{a_i}{A_t} \langle \nabla f(x_i), v - x_i \rangle$, and we refer to this lower bound as $L_t^{\text{HB}}$ if not clear from the context. Clearly this is a lower bound and in fact this is precisely the heavy ball vertex that is computed in the heavy ball algorithm. It is important to note that this lower bound and the associated dual gap is valid for *any algorithm* as it is simply a convex combination of linear lower bound functions minimized over $P$. For comparability and due to its natural form and being valid for any algorithm, we use this definition of the gap also for reporting the results of the optimistic variant in our experiments.

In the case of the vanilla FW variant (either with open loop step sizes or with line search) the $v_i$ in the expression above are chosen to be the FW vertices, i.e., $v_i \in \arg\min_{v \in P} f(x_i) + \langle \nabla f(x_i), v - x_i \rangle$, minimizing each summand separately. This bound is thus separable in contrast to the heavy ball one as we will discuss further below in Remark 5.1 and we can use $f(x^*) \ge \max_{0 \le i \le t} f(x_i) + \langle \nabla f(x_i), v_i - x_i \rangle$, as lower bound so that the gap becomes the running minimum of the FW gaps across the iterations, which we refer to as $L_t^{\text{FW}}$.

**Remark 5.1** (Strength of lower bounds). *Suppose that $g(x_t)$ denotes a generic gap function that bounds the primal gap at $x_t$, i.e., $f(x_t) - f(x^*) \le g(x_t)$. We will choose the specific gap function later depending on the context. Observe that the gap function immediately gives a lower bound for $f(x^*)$ simply by rewriting as $f(x^*) \ge f(x_t) - g(x_t)$. Then in line with the above if we make the choice for the lower bound $L_t$ as $A_t f(x^*) \ge \sum_{\ell=0}^{t} a_\ell f(x_\ell) - \sum_{\ell=0}^{t} a_\ell g(x_\ell) = A_t L_t$, then in this case, the lower bound $L_t$ cannot be stronger than taking the best lower bound observed so far, since $f(x^*) \ge \max_{\ell=0,\dots,t} f(x_\ell) - g(x_\ell) \ge \frac{1}{A_t} \sum_{\ell=0}^{t} a_\ell (f(x_\ell) - g(x_\ell))$, This is the case because the lower bound is a convex combination of lower bound terms from individual iterations.*

*As mentioned, this is different, e.g., for the heavy ball lower bound function, which does not decompose in individual iterations as the Frank-Wolfe vertex is computed for the cumulative function across rounds. The reason why the primal-dual gap for HB-FW can be significantly better (as in: lower)*

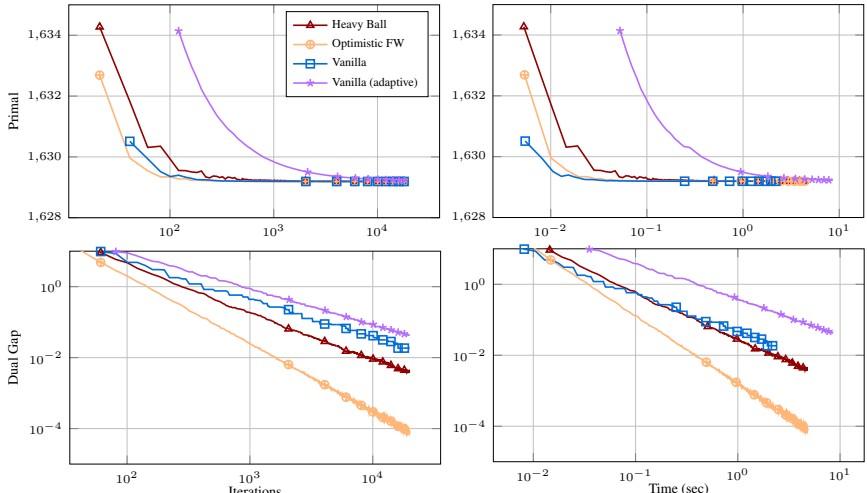

Figure 2: Comparison over $k$-sparse polytope with $k = 10$ with objective $f(x) = \|Ax - b\|_2^2$, where $A$ and $b$ are random. The optimistic method converges faster in iterations and time.

*than the Frank-Wolfe gap is illustrated in the following example: The FW algorithm, commonly used with polytopal constraints, sometimes suffers from the so-called zigzag problem (Wolfe, 1970; Guélat & Marcotte, 1986) (see also Braun et al. (2022) for an in-depth discussion), that usually is due to a minimizer being in the relative interior of a face while the points $v_t$, being the result of an LMO, are chosen as vertices of the polytope. In this scenario, it is possible that $\nabla f(x_t)$ is aligned with the direction $v_t - x_t$ for all $t$, while a convex combination of gradients is close to being perpendicular to the optimal face making the lower bound $L_t^{\mathrm{HB}}$ be closer to the optimal value.*

In the plots we show the primal value and the dual gap vs. the number of iterations and time. Points with excessively large values are not shown in the plots, leading to apparent different starting points. We use log-log plots so that the slope is equal to the polynomial order of convergence. As our variants compute one gradient per iteration, the number of iterations is equal to the number of gradient evaluations. Throughout our experiments we see that the optimistic variant significantly outperforms the heavyball variant, the vanilla FW, and often, although not always, also the FW variant with the adaptive line search strategy from (Pokutta, 2024) in the order of convergence.

We also run experiments for the primal-dual short step from Section 4. However, we found that it behaved similarly than the vanilla FW variant with standard short-steps (or line search), without outperforming this already good heuristic; see Figure 3 in Appendix F. The reason seems to be that because the primal-dual gap measure is better, the step size is actually often *smaller*, leading to no faster convergence; this is reminiscent of the lower bound in (Guélat & Marcotte, 1986).

We also tested whether optimism is really the main source of improvement or whether most of it is explained by using the heavy ball lower bound which shows to be stronger than the vanilla FW lower bound. In fact, we show in Figure 4 in Appendix F, that indeed optimism is what is making the algorithm be faster. In fact the heavy ball lower bound applied to the vanilla FW variant is weaker that the normal FW gap in our tests. This strongly suggests that it is really the trajectory of the optimistic variant that is better. For more experiments, we refer to Appendix G.

We tested various combinations of functions and feasible regions. For the functions, we used (a) quadratic optimization consisting essentially on (b) ill-conditioned quadratic optimization and (c) portfolio optimization problems, which are all widespread benchmarks for Frank-Wolfe algorithms (Braun et al., 2022), see Appendix F. In terms of the feasible regions, we considered (i) the probability simplex $P = \{x \in \mathbb{R}^d \mid \sum_i x_i = 1, x \geq 0\}$, (ii) and the more complex $k$-sparse polytope $P$ which is the convex hull of all $0/1$-vectors with $k$ non-zero entries. We detail the instance parameters in the captions of the figures.

In our experiments we report results for regression problems. The reason for this is to allow us to build specific setups where the different convergence behaviors of FW variants are known to emerge. We also performed preliminary experiments over portfolio optimization instances (here the objective is the negative log-likelihood of the portfolio returns) and also Optimal Experiment Design instances (Hendrych et al., 2024) (here the objective arises via matrix means) and found that the optimistic variant also outperforms the other variants similarly.

The optimistic approach presents a good alternative for current Frank-Wolfe methods that is theoretically justified and enjoys faster convergence in practice.

ACKNOWLEDGEMENTS

Research reported in this paper was partially supported through the Research Campus Modal funded by the German Federal Ministry of Education and Research (fund numbers 05M14ZAM,05M20ZBM) and the Deutsche Forschungsgemeinschaft (DFG) through the DFG Cluster of Excellence MATH+ (EXC-2046/1 and EXC-2046/2, project id 390685689). David Martínez-Rubio was partially funded by the project IDEA-CM (TEC-2024/COM-89).

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
