## A    USE OF GENERATIVE AI

Generative AI was used for polishing the writing of this paper. No other use of generative AI was made for this work.

## B    OVERVIEW AND SOME GENERALIZATIONS OF FRANK-WOLFE'S PRIMAL-DUAL ANALYSES

As discussed in the introduction, the analysis of Frank-Wolfe algorithms on compact convex sets can be generalized to the problem $f + \psi$ where we assume access to a gradient oracle for $f$ and an that solves the global optimization of the function $\arg\min_{x \in \mathbb{R}^d} \langle w, x \rangle + \psi(x)$ for every $x$, as originally proposed in (Nesterov, 2018). Here, we provide a short proof of this fact, with a slightly better constant, which in particular proves the claims in Section 2 when $\psi$.

Assume $f : \mathbb{R}^n \to \mathbb{R}$ is convex, $L$-smooth, and differentiable in an open set containing $\mathcal{X} \overset{\text{def}}{=} \text{dom}(\psi)$ and $\psi : \mathbb{R}^n \to R$ is a convex, proper, lower semi-continuous function. We assume that for every $u \in \mathbb{R}^n$, there exists $\arg\min_x \{\langle u, x \rangle + \psi(x)\}$. Finally, the convergence after $t$ steps will depend on $D_t \overset{\text{def}}{=} \max_{\ell=0,\ldots,t}\{\|v_\ell - x_\ell\|\}$ for the points $x_\ell, v_\ell$ with $\ell = 0, \ldots, t$ as defined in Algorithm 1 up to iteration $t$. Similar to the original FW algorithm, we can instead substitute $D_t$ by an upper bound of such quantity, e.g., $\text{diam}(\text{dom}(\psi))$, assuming its value is finite. Recall that we define positive weights $a_\ell > 0$ for all $\ell \geq 0$, to be determined later, and $A_t \overset{\text{def}}{=} \sum_{i=0}^t a_i$. Thus, $A_{-1} = 0$.

---

**Algorithm 3** Generalized Frank-Wolfe and Heavy-Ball FW algorithms

**Input:** Functions $f$ and $\psi$, initial point $x_0$. Weights $a_t, \gamma_t$.
1: **for** $t = 0$ **to** $T$ **do**
2:     $a_t \leftarrow 2t + 2$, $A_t \leftarrow \sum_{i=0}^t a_i = (t+1)(t+2)$
3:     $\diamond$ Choose either 3 (generalized HB-FW) or 4 (generalized FW) for the entire run:
4:     $v_t \leftarrow \underset{v \in \mathbb{R}^d}{\arg\min} \left\{ \sum_{i=0}^t a_i \langle \nabla f(x_i), v \rangle + A_t \psi(v) \right\}$
5:     $v_t \leftarrow \underset{v \in \mathbb{R}^d}{\arg\min} \left\{ \langle \nabla f(x_t), v \rangle + \psi(v) \right\}$
6:     $x_{t+1} \leftarrow \frac{A_{t-1}}{A_t} x_t + \frac{a_t}{A_t} v_t$
7: **end for**

---

We define the following lower bound on $f(x^*) + \psi(x^*)$, which is a generalization of (2), where $x^*$ here is defined as a minimizer in $\arg\min_{x \in \mathbb{R}^n}\{f(x) + \psi(x)\}$:

$$A_t(f(x^*) + \psi(x^*)) \overset{\text{①}}{\geq} \sum_{\ell=0}^t a_\ell f(x_\ell) + \sum_{\ell=0}^t a_\ell \langle \nabla f(x_\ell), x^* - x_\ell \rangle + \sum_{\ell=0}^t a_\ell \psi(x^*)$$

$$\overset{\text{②}}{\geq} \sum_{\ell=0}^t a_\ell f(x_\ell) + \sum_{\ell=0}^t a_\ell \langle \nabla f(x_\ell), v_\ell - x_\ell \rangle + \sum_{\ell=0}^t a_\ell \psi(v_\ell)$$

$$\overset{\text{def}}{=} A_t L_t,$$

where we applied convexity in ①, and for ②, we applied the definition of $v_\ell$ in the algorithm, which is why we define this point. Define the primal-dual gap

$$G_t \overset{\text{def}}{=} f(x_{t+1}) - L_t \tag{7}$$

and note that upper bound $f(x_{t+1})$ is one step ahead, which helps the analysis. We obtain the following, for $t \geq 0$:

$$A_t G_t - A_{t-1} G_{t-1} = A_t f(x_{t+1}) - A_{t-1} f(x_t) + A_t \psi(x_{t+1}) - A_{t-1} \psi(x_t)$$

$$- \left( a_t f(x_t) + \sum_{\ell=0}^{t-1} a_\ell f(x_\ell) + a_t \langle \nabla f(x_t), v_t - x_t \rangle + a_t \psi(v_t) + \sum_{\ell=0}^{t-1} a_\ell (\langle \nabla f(x_\ell), v_\ell - x_\ell \rangle + \psi(v_\ell)) \right)$$

$$+ \left( \sum_{\ell=0}^{t-1} a_\ell f(x_\ell) + \sum_{\ell=0}^{t-1} a_\ell (\langle \nabla f(x_\ell), v_\ell - x_\ell \rangle + \psi(v_\ell)) \right)$$

$$\overset{①}{\leq} \langle \nabla f(x_t), A_t(x_{t+1} - x_t) - a_t(v_t - x_t) \rangle + \frac{L A_t}{2} \|x_{t+1} - x_t\|^2$$

$$\overset{②}{=} \frac{L a_t^2}{2 A_t} \|v_t - x_t\|^2 \overset{③}{\leq} \frac{L D_t^2 a_t^2}{2 A_t} \overset{\text{def}}{=} E_t.$$

$$(8)$$

In ① we grouped terms to get $A_t(f(x_{t+1}) - f(x_t))$ and applied smoothness to this term. We also used the definition of $x_{t+1}$ which along with the convexity of $\psi$, implies $A_t \psi(x_{t+1}) \leq A_{t-1} \psi(x_t) + a_t \psi(v_t)$. In ② we used twice the definition of $x_{t+1}$ which implies $A_t(x_{t+1} - x_t) = a_t(v_t - x_t)$. In ③, we bounded the distance of points by the quantity $D_t$.

Now with the choice $a_t = 2t + 2$ and $A_t = \sum_{i=0}^{t} a_i = (t+1)(t+2)$ that

$$f(x_{t+1}) + \psi(x_{t+1}) - (f(x^*) + \psi(x^*)) \leq G_t$$

$$\leq \frac{1}{A_t} \sum_{i=0}^{t} \frac{L D_t^2 a_i^2}{2 A_i} = \frac{1}{(t+1)(t+2)} \sum_{i=0}^{t} \frac{2 L D_t^2 (i+1)}{i+2} < \frac{2 L D_t^2}{t+2}. \qquad (9)$$

Recall that the classical Frank-Wolfe algorithm corresponds to the algorithm and analysis presented above but for $\psi$ being the indicator function $I_{\mathcal{X}}$ of a compact convex set $\mathcal{X}$. In such a case, we have the following.

**Remark B.1** (Alternative step-sizes in Generalized Frank-Wolfe). *We note that step-size strategies like the ones described in Remark 2.1 also work in the general case $f + \psi$, by finding*

$$\underset{x \in \text{conv } \hat{x}_t, v_t}{\arg\min} \left\{ f(\hat{x}_t) + \langle \nabla f(\hat{x}_t), x - \hat{x}_t \rangle + \frac{L}{2} \|x - \hat{x}_t\|^2 + \psi(x) \right\} \quad or \quad \underset{x \in \text{conv } \hat{x}_t, v_t}{\arg\min} \{f(x) + \psi(x)\},$$

*respectively.*

## B.1 THE HEAVY BALL FRANK-WOLFE ALGORITHM

We will now consider a variant of the Frank-Wolfe algorithm that uses a heavy ball step, similar to the one from Li et al. (2021), however allowing for more flexibility in the choice of the step size parameters and an additional term $\psi(v)$. As before, let $a_i > 0$ to be determined later and define $A_t = \sum_{i=0}^{t} a_i$. Let $v_t \overset{\text{def}}{\in} \arg\min_{v \in \mathcal{X}} \left\{ \langle \sum_{i=0}^{t} a_i \nabla f(x_i), v \rangle + A_t \psi(v) \right\}$ and let $x_{t+1} \overset{\text{def}}{=} \frac{A_{t-1}}{A_t} x_t + \frac{a_t}{A_t} v_t$ be defined as a convex combination of $x_t$ and $v_t$ (and since $A_0 = a_0$, we have $x_1 = v_0$). We define the following lower bound on $f(x^*)$, where $x^*$ is defined as a minimizer in $\arg\min_{x \in \mathcal{X}} \{f(x) + \psi(x)\}$:

$$A_t(f(x^*) + \psi(x^*)) \overset{①}{\geq} \sum_{i=0}^{t} a_i f(x_i) + \sum_{i=0}^{t} a_i \langle \nabla f(x_i), x^* - x_i \rangle + A_t \psi(x^*)$$

$$\overset{②}{\geq} \sum_{i=0}^{t} a_i f(x_i) + \sum_{i=0}^{t} a_i \langle \nabla f(x_i), v_t - x_i \rangle + A_t \psi(v_t) \overset{\text{def}}{=} A_t L_t$$

where we applied convexity in ①, and the definition of $v_t$ in ②. As before, we define the primal-dual gap as

$$G_t \overset{\text{def}}{=} f(x_{t+1}) - L_t. \qquad (10)$$

We obtain the following, for $t \geq 0$:

$$A_t G_t - A_{t-1} G_{t-1} = A_t f(x_{t+1}) - A_{t-1} f(x_t) + A_t \psi(x_{t+1}) - A_{t-1} \psi(x_t)$$

$$- \left( a_t f(x_t) + \sum_{i=0}^{t-1} a_i f(x_i) + \sum_{i=0}^{t-1} a_i \langle \nabla f(x_i), v_t - x_i \rangle + A_{t-1} \psi(v_t) \right) - a_t \langle \nabla f(x_t), v_t - x_t \rangle - a_t \psi(v_t)$$

$$+ \left( \sum_{i=0}^{t-1} a_i f(x_i) + \sum_{i=0}^{t-1} a_i \langle \nabla f(x_i), v_{t-1} - x_i \rangle + A_{t-1} \psi(v_{t-1}) \right)$$

$$\overset{\text{①}}{\leq} \langle \nabla f(x_t), A_t(x_{t+1} - x_t) - a_t(v_t - x_t) \rangle + \frac{LA_t}{2} \|x_{t+1} - x_t\|^2$$

$$\overset{\text{②}}{\leq} \frac{La_t^2}{2A_t} \|v_t - x_t\|^2 \overset{\text{③}}{\leq} \frac{LD_t^2 a_t^2}{2A_t} \overset{\text{def}}{=} E_t.$$

$$(11)$$

In ① we grouped terms to get $A_t(f(x_{t+1}) - f(x_t))$ and applied smoothness to this term. We also dropped the rest of the terms in the parentheses by optimality of of $v_{t-1}$ and dropped the three other terms depending on $\psi$ by its convexity and $A_t x_{t+1} = A_{t-1} x_t + a_t v_t$. In ② we used the definition of $x_{t+1}$, which is the point that minimizes the right hand side of the smoothness inequality that we applied and so substituting $x_{t+1}$ by $\frac{A_{t-1}}{A_t} x_t + \frac{a_t}{A_t} v_t$ leads to something greater. We simplified some terms. In ③, we bounded the distance of points by the diameter of $\mathcal{X}$. Concluding is now the same as for FW.

## B.2 Generalized Frank-Wolfe algorithm revisited

In (8) we have obtained a convergence rate for $\min_x f(x) + \psi(x)$. Alternatively, if we are interested in optimizing $f$, we can use a regularizer $\psi$ and conclude convergence with a very similar analysis. As a consequence, $\psi(x^*)$ appears in the rates. If $\psi$ is an indicator function of a set this is exactly the heavy ball algorithm from above but for a general $\psi$, it is different.

The lower bound that we define on $f(x^*)$ with $x^* \in \arg\min_{x \in \mathcal{X}} f(x)$, and $\psi$ as before, is as follows:

$$A_t f(x^*) \overset{\text{①}}{\geq} \sum_{i=0}^{t} a_i f(x_i) + \sum_{i=0}^{t} a_i \langle \nabla f(x_i), x^* - x_i \rangle \pm \psi(x^*)$$

$$\geq \sum_{i=0}^{t} a_i f(x_i) + \sum_{i=0}^{t} a_i \langle \nabla f(x_i), v_t - x_i \rangle + \psi(v_t) - \psi(x^*) \overset{\text{def}}{=} A_t L_t,$$

$$(12)$$

where $v_t \overset{\text{def}}{\in} \arg\min_{v \in \mathbb{R}^n} \{ \sum_{i=0}^{t} a_i \langle \nabla f(x_i), v \rangle + \psi(v) \}$. Similarly to the previous section, we assume access to an oracle that returns one such $v_t$, which is assumed to exist. We used convexity in ① and we also added and subtracted $\psi$ in order to compute a lower bound that allows us to reduce

the gap without knowing $x^*$. Finally, defining the gap $G_t \stackrel{\text{def}}{=} f(x_{t+1}) - L_t$, we have, for all $t \geq 0$

$$
\begin{aligned}
A_t G_t - A_{t-1} G_{t-1} - &1_{\{t=0\}} \psi(x^*) = A_t f(x_{t+1}) - A_{t-1} f(x_t) \\
&- \left( a_t f(x_t) + \sum_{i=0}^{t-1} a_i f(x_i) + a_t \langle \nabla f(x_t), v_t - x_t \rangle + \sum_{i=0}^{t-1} a_i \langle \nabla f(x_i), v_t - x_i \rangle + \psi(v_t) \right) \\
&+ \left( \sum_{i=0}^{t-1} a_i f(x_i) + \sum_{i=0}^{t-1} a_i \langle \nabla f(x_i), v_{t-1} - x_i \rangle + \psi(v_{t-1}) \right) \\
\stackrel{①}{\leq} & \langle \nabla f(x_t), A_t(x_{t+1} - x_t) - a_t(v_t - x_t) \rangle + \frac{LA_t}{2} \|x_{t+1} - x_t\|^2 \\
\stackrel{②}{=} & \frac{La_t^2}{2A_t} \|v_t - x_t\|^2 \stackrel{③}{\leq} \frac{LD^2 a_t^2}{2A_t} \stackrel{\text{def}}{=} E_t.
\end{aligned}
$$

(13)

Step ① uses the optimality of $v_{t-1}$ to bound some terms by $0$, Steps ② and ③ are identical to the previous analysis. Now to conclude we choose $a_t = 2t + 2$ as before, and so we have $A_t = \sum_{i=0}^{t} a_i = (t+1)(t+2)$. But we also have to take into account that we have the extra term $\psi(x^*)$ when bounding $A_0 G_0$:

$$
\begin{aligned}
f(x_{t+1}) - f(x^*) &\leq \frac{1}{A_t} \left( \psi(x^*) + \sum_{i=0}^{t} E_i \right) = \frac{\psi(x^*)}{A_t} + \frac{1}{A_t} \sum_{i=0}^{t} \frac{LD^2 a_i^2}{2A_i} \\
&= \frac{\psi(x^*)}{A_t} + \frac{1}{(t+1)(t+2)} \sum_{i=0}^{t} \frac{2LD^2(i+1)}{(i+2)} < \frac{\psi(x^*)}{(t+1)(t+2)} + \frac{2LD^2}{t+2}.
\end{aligned}
$$

**Remark B.2** (Arbitrary fast rate for $\psi(x^*)$). *In fact, the part of the rate involving $\psi(x^*)$ was arbitrary and made for simplicity. Indeed, the intuition is that in the lower bound* (12) *we are adding a regularizer that is in fact $\psi(x^*)/A_t$ (since the whole inequality is multiplied by $A_t$ in particular $f(x^*)$), so the larger we make $A_t$ be, the smaller the regularizer we are adding is, and the faster that term will go to $0$. In algebra, if we set $a_i = \Theta(i^k)$, then we have that $\sum_{i=0}^{t} a_i^2/A_i = \Theta(\sum_{i=0}^{t} i^{2k-(k+1)}) = \Theta(t^k)$ and $A_t = \Theta(t^{k+1})$ so in any case the part without regularizer is*

$$
LD^2 \frac{1}{A_t} \sum_{i=0}^{t} \frac{a_i^2}{A_i} = \Theta \left( \frac{LD^2}{t} \right),
$$

*but now $A_t$ grows at any polynomial rate that we want which we can use to decrease $\psi(x^*)/A_t$ fast. If we are adding a $\psi$ ourselves on top of an indicator function because we want some properties to hold, most of the time we would like to keep this term being of the same order as the current gap so its contribution to the total rate of convergence is a multiplicative constant.*

*This reasoning also applies when finishing the analysis of Theorem 3.1.*

## C PROOFS FOR OPTIMISTIC FW ALGORITHM

We start by defining the lower bound that we use on $f(x^*)$ for a family of algorithms, using which we define a primal-dual gap that allows to show convergence of our optimistic Frank-Wolfe algorithm. This proof is inspired by the anytime online-to-batch reduction of an optimization problem into an online learning one (Cutkosky, 2019), which is a generalization and independent work with respect to (Nesterov & Shikhman, 2015). An interesting modification of the technique that we make is that we allow for selecting a point $y_{t-1}$ such that $f(y_{t-1}) \leq f(x_{t-1})$ before computing the coupling $x_t$. This allows to incorporate heuristics without degrading the convergence rate, such as performing a line search between $x_{t-1}$ and $v_t$. We used this modified reduction in order to provide a primal-dual gap, which is computable if the regret of the corresponding online learning problem is computable. If we instead have a computable upper bound $\widehat{R}_t(x^*)$ on the regret $R_t(x^*)$ we can correspondingly

define and compute a primal-dual gap based on this bound. This will be the case when we instantiate the framework with our optimistic algorithm.

Let $a_t > 0$ for $t \geq 1$ and define $A_t \overset{\text{def}}{=} \sum_{i=1}^{t} a_i$. Given some points $\{v_i\}_{i\geq 1}$, and given $x_0$, we define $x_t \overset{\text{def}}{=} \frac{A_{t-1}}{A_t} y_{t-1} + \frac{a_t}{A_t} v_t$, for $t \geq 1$, where $y_t$ is any point such that $f(y_t) \leq f(x_t)$. Then, we have for all $t \geq 1$ and $k = 0, 1, \ldots, t$:

$$
\begin{aligned}
A_t f(x^*) &\overset{\text{①}}{\geq} \sum_{i=1}^{t} a_i f(x_i) + \sum_{i=1}^{t} a_i \langle \nabla f(x_i), x^* - x_i \rangle \\
&\overset{\text{②}}{=} \Phi_k \overset{\text{def}}{=} A_k f(x_k) + \sum_{i=1}^{k} a_i \langle \nabla f(x_i), x^* - v_i \rangle + \sum_{i=k+1}^{t} a_i f(x_i) + \sum_{i=k+1}^{t} a_i \langle \nabla f(x_i), x^* - x_i \rangle \\
&\quad + \sum_{i=0}^{k-1} \Big( A_i D_f(y_i, x_{i+1}) + A_i(f(x_i) - f(y_i)) \Big) \\
&\overset{\text{③}}{=} A_t f(x_t) + \sum_{i=1}^{t} a_i \langle \nabla f(x_i), x^* - v_i \rangle + \sum_{i=0}^{t-1} \Big( A_i D_f(y_i, x_{i+1}) + A_i(f(x_i) - f(y_i)) \Big) \\
&\overset{\text{④}}{\geq} A_t f(x_t) - \widehat{R}_t(x^*) \overset{\text{def}}{=} A_t L_t
\end{aligned}
\tag{14}
$$

where ① uses convexity of $f$. We have that ② and ③ are due to the terms on the sides being $\Phi_0$ and $\Phi_t$ and to $\Phi_k = \Phi_{k+1}$, which holds, since canceling terms such equality is equivalent to ⑤ below:

$$
\begin{aligned}
A_k f(x_k) &\overset{\text{⑤}}{=} a_{k+1} \langle \nabla f(x_{k+1}), x_{k+1} - v_{k+1} \rangle + A_k f(x_{k+1}) + A_k D_f(y_k, x_{k+1}) + A_k(f(x_k) - f(y_k)) \\
&\overset{\text{⑥}}{=} A_k \langle \nabla f(x_{k+1}), y_k - x_{k+1} \rangle + A_k f(x_{k+1}) + A_k D_f(y_k, x_{k+1}) + A_k(f(x_k) - f(y_k)),
\end{aligned}
$$

where ⑥ holds by definition of $x_{k+1}$. Thus, ⑤ clearly holds since the right hand side of ⑥ equals the left hand side of ⑤, by definition of the Bregman divergence. Note that the second summand of the right hand side of ③ equals to minus the regret $R_t(x^*)$ of the online learning game with linear losses $a_i \langle \nabla f(x_t), \cdot \rangle$ and played points $v_i$ with respect to the comparator $x^*$. We defined $\widehat{R}_t(x^*)$ as any computable upper bound on $R_t(x^*)$, and thus we obtain ④ above by this bound, the assumption on $y_{t-1}$, and $D_f(y_i, x_{i+1}) \geq 0$.

Now, we define our primal-dual gap as

$$
G_t \overset{\text{def}}{=} f(x_t) - L_t = \frac{\widehat{R}_t(x^*)}{A_t},
\tag{15}
$$

which, by construction, is an upper bound on the primal gap $f(x_t) - f(x^*)$. Thus, an online learning algorithm whose regret, or a bound of it, is computable, provides us with a computable primal-dual gap, and convergence is linked to the value of the regret. We can now apply optimistic follow-the-regularized leader (OFTRL) or optimistic Mirror Descent (OMD) online learning algorithms and provide regret bounds, by using a not necessarily strongly convex or differentiable regularizer in order to obtain an optimistic generalized FW algorithm, where we assume that we can solve subproblems of the form $\arg\min_x \{\langle w, x \rangle + \psi(x)\}$, for any $w \in \mathbb{R}^d$. Typically in FW algorithms, $\psi$ is just the indicator function of the feasible set. We provide an overview of these algorithms, that we have generalized to deal with subdifferentiable regularizers.

## C.1 Optimistic FTRL and optimistic MD with convex subdifferentiable regularizers

Given a function $\psi$ that is subdifferentiable in a closed convex feasible set $\mathcal{X}$, two points $x, y \in \mathcal{X}$ and $\phi \in \partial\psi(y)$, define the non-differentiable Bregman divergence $D_\psi(x, y, \phi)$ as

$$
D_\psi(x, y; \phi) \overset{\text{def}}{=} \psi(x) - \psi(y) - \langle \phi, x - y \rangle,
$$

We start by presenting a regret bound for Optimistic FTRL, by using our possibly non-differentiable non-strongly convex regularizers.

**Theorem C.1** (Optimistic FTRL). *Let $\mathcal{X}$ be a closed convex set and let $\psi_t, \ell_t : \mathcal{X} \to \mathbb{R}$ be closed, proper, convex and subdifferentiable functions in $\mathcal{X}$, for $t \geq 1$. For $t \in [T]$ and given some hints $\tilde{g}_t \in \mathbb{R}^d$, let $z_t \stackrel{\text{def}}{\in} \arg\min_{z \in \mathcal{X}} \{\sum_{i=1}^{t-1} \ell_i(z) + \langle \tilde{g}_t, z \rangle + \psi_t(z)\}$, which we assume to exist. Also, define $z_{T+1} = u$ be an arbitrary point $u \in \mathcal{X}$. Then, the regret $R_T(u)$ satisfies:*

$$\sum_{t=1}^{T} (\ell_t(z_t) - \ell_t(u)) \leq \psi_{T+1}(u) - \min_{z \in \mathcal{X}} \psi_1(z) + \sum_{t=1}^{T} \left( \langle g_t - \tilde{g}_t, z_t - z_{t+1} \rangle - D_{f_t}(z_{t+1}, z_t; g_t - \tilde{g}_t) + \psi_t(z_{t+1}) - \psi_{t+1}(z_{t+1}) \right),$$

*for all subgradients $g_t \in \partial \ell_t(z_t)$, where $f_t(z) \stackrel{\text{def}}{=} \sum_{i=1}^{t} \ell_i(z) + \psi_t(z)$.*

*Proof.* First, note that since $z_t = \arg\min_{z \in \mathcal{X}} \{f_t(z) - \ell_t(z) + \langle \tilde{g}_t, z \rangle\}$, we have $0 \in \partial(f_t - \ell_t + \langle \tilde{g}_t, \cdot \rangle)(z_t)$ and thus $g_t - \tilde{g}_t \in \partial f(z_t)$ for any $g_t \in \partial \ell_t(z_t)$, so the expression $D_{f_t}(z_{t+1}, z_t; g_t - \tilde{g}_t)$ above makes sense. We bound the regret as

$$\sum_{t=1}^{T} (\ell_t(z_t) - \ell_t(u)) = \sum_{t=1}^{T} \left[ \left( \ell_t(z_t) + \sum_{i=1}^{t-1} \ell_i(z_t) + \psi_t(z_t) \right) - \left( \sum_{i=1}^{t} \ell_i(z_{t+1}) + \psi_{t+1}(z_{t+1}) \right) \right]$$
$$+ \psi_{T+1}(u) - \psi_1(z_1)$$
$$= \sum_{t=1}^{T} \left[ \left( \sum_{i=1}^{t} \ell_i(z_t) + \psi_t(z_t) \right) - \left( \sum_{i=1}^{t} \ell_i(z_{t+1}) + \psi_t(z_{t+1}) \right) + \psi_t(z_{t+1}) - \psi_{t+1}(z_{t+1}) \right]$$
$$+ \psi_{T+1}(u) - \psi_1(z_1)$$
$$\leq \psi_{T+1}(u) - \min_{z \in \mathcal{X}} \psi_1(z) + \sum_{t=1}^{T} \left( \langle g_t - \tilde{g}_t, z_t - z_{t+1} \rangle - D_{f_t}(z_{t+1}, z_t; g_t - \tilde{g}_t) + \psi_t(z_{t+1}) - \psi_{t+1}(z_{t+1}) \right).$$
$$(16)$$

Above, we simply add and subtract terms, use the definition of $z_{T+1}$, and in the inequality we just bound $-\psi_1(z_1) \leq -\min_{z \in \mathcal{X}} \psi_1(z)$. $\qquad \square$

**Corollary C.2.** *Under the assumptions of Theorem C.1, for time-invariant $\psi_t = \psi$ we have*

$$\sum_{t=1}^{T} (\ell_t(z_t) - \ell_t(u)) \leq \psi_{T+1}(u) - \min_{z \in \mathcal{X}} \psi_1(z) + \sum_{t=1}^{T} \langle g_t - \tilde{g}_t, z_t - z_{t+1} \rangle - D_{f_t}(z_{t+1}, z_t; g_t - \tilde{g}_t)$$
$$\leq \psi_{T+1}(u) - \min_{z \in \mathcal{X}} \psi_1(z) + \sum_{t=1}^{T} \langle g_t - \tilde{g}_t, z_t - z_{t+1} \rangle,$$
$$(17)$$

*Proof.* The result follows by noticing that the terms with $\psi_t$ in the sum cancel and that the Bregman divergences of the convex functions $f_t$ are non-negative. $\qquad \square$

Now we present an alternative algorithm to the above, the optimistic Mirror Descent algorithm. First, we prove a lemma about the generic update rule of Mirror Descent: $x_{t+1} \in \arg\min_{x \in X} \{\eta_t \langle g, x \rangle + D_\psi(x, x_t, \phi_t)\}$, where an assumption is made for $\psi$ that the $\arg\min$ is always non empty. This holds for instance, for $\psi$ being the indicator function of a convex compact set, or $\psi$ being a Legendre function. The following corresponds to the classical mirror descent lemma, but for non-differentiable maps.

**Lemma C.3** (Mirror Lemma with non-Differentiable Mirror Map). *Given a closed convex set $\mathcal{X}$, let $\psi$ be proper, closed, convex and subdifferentiable in $\mathcal{X}$, and let $g \in \mathbb{R}^d$. If $x_{t+1} \in \arg\min_{x \in \mathcal{X}} \{\langle g, x \rangle + D_\psi(x, x_t; \phi_t)\}$ exists, then for all $u \in \mathcal{X}$ and all $\phi_{t+1} \in \partial \psi(x_{t+1})$:*

$$\langle g, x_{t+1} - u \rangle \leq D_\psi(u, x_t; \phi_t) - D_\psi(x_{t+1}, x_t; \phi_t) - D_\psi(u, x_{t+1}; \phi_{t+1}).$$

We note that by optimality of $x_{t+1}$, we have $0 \in g + \partial \psi(x_{t+1}) - \phi_t$ and so a possible choice of $\phi_{t+1}$ is $\phi_t - g$.

*Proof.* The point $x_{t+1}$ is a minimizer of $F(x) \stackrel{\text{def}}{=} \langle g, x \rangle + D_\psi(x, x_t; \phi_t)$. If we substitute the definition of $F$ into the following expression, implied by the first order optimality condition of $x_{t+1}$, $F(x_{t+1}) + D_F(u, x_{t+1}; \phi_{t+1}) \leq F(u)$, then we obtain the lemma above. $\qquad\square$

The following theorem about mirror descent is a slight generalization over the common one using not necessarily differentiable regularizers and using a hint for the first step. Compare with, for instance, (Orabona, 2019, Theorem 6.20) with constant step size.

**Theorem C.4** (Optimistic MD). *Let $\mathcal{X}$ be a closed convex set and let $\psi, \ell_t : \mathcal{X} \to \mathbb{R}$ be closed, proper, convex and subdifferentiable functions in $\mathcal{X}$, for $t \in [T]$. For $t \in [T]$, and given some hints $\tilde{g}_t \in \mathbb{R}^d$ and $g_0 = \tilde{g}_0 = 0$, $z_0 \in \mathcal{X}$, let $z_t \stackrel{\text{def}}{\in} \arg\min_{z \in \mathcal{X}}\{\langle g_{t-1} + \tilde{g}_t - \tilde{g}_{t-1}, z \rangle + D_\psi(z, z_{t-1}; \phi_{t-1})\}$ for $\phi_{t-1} \in \partial\psi(z_{t-1})$, which we assume it exists. Also define $z_{T+1} = u$ as an arbitrary point $u \in \mathcal{X}$. Then, the regret $R_T(u)$ satisfies:*

$$\sum_{t=1}^{T} (\ell_t(z_t) - \ell_t(u)) \leq D_\psi(u, z_0; \phi_0) + \sum_{t=1}^{T} \Big( \langle g_t - \tilde{g}_t, z_t - z_{t+1} \rangle - D_\psi(z_{t+1}, z_t; \phi_t) \Big) - D_\psi(z_1, z_0; \phi_0)$$

$$\leq D_\psi(u, z_0; \phi_0) + \sum_{t=1}^{T} \langle g_t - \tilde{g}_t, z_t - z_{t+1} \rangle. \tag{18}$$

*for all subgradients $g_t \in \partial\ell_t(z_t)$, $t \geq 1$, where $z_{T+1}$ is defined as above with $\tilde{g}_{T+1} = 0$.*

*Proof.* Fix the choices $g_t \in \partial\ell_t(z_t)$ for $t \in [T]$ and define $\phi_t \stackrel{\text{def}}{=} g_{t-1} - \tilde{g}_t + \tilde{g}_{t+1}$. Applying Lemma C.3 and adding a term with $z_t$, we obtain

$$\langle g_t + \tilde{g}_{t+1} - \tilde{g}_t, z_t - u \rangle \leq D_\psi(u, z_t; \phi_t) - D_\psi(z_{t+1}, z_t, \phi_t) - D_\psi(u, z_{t+1}; \phi_{t+1}) + \langle g_t + \tilde{g}_{t+1} - \tilde{g}_t, z_t - z_{t+1} \rangle. \tag{19}$$

Adding up the above from $t = 0$ to $T$, and taking into account that $g_0 = \tilde{g}_0 = 0$, and setting $\tilde{g}_{T+1}$ we obtain an inequality whose left hand side is

$$\sum_{t=0}^{T} \langle g_t + \tilde{g}_{t+1} - \tilde{g}_t, z_t - u \rangle = \sum_{t=1}^{T} \langle g_t, z_t - u \rangle + \langle \tilde{g}_{T+1} - \tilde{g}_0, u \rangle + \sum_{t=0}^{T} \langle \tilde{g}_{t+1} - \tilde{g}_t, z_t \rangle$$

$$\overset{\text{①}}{\geq} \sum_{t=1}^{T} \Big( \ell_t(z_t) - \ell_t(u) \Big) + \sum_{t=0}^{T} \langle \tilde{g}_{t+1}, z_t - z_{t+1} \rangle.$$

Note that we can set $\tilde{g}_{T+1}$ to any value since it does not play a role in any of the first $T$ predictions. For simplicity, we used $\tilde{g}_{T+1} = 0$. In ① above, for the first summand we used the subgradient property, the second summand vanished by our choice of hints and then we used an equality for the last term, using again that $\tilde{g}_0 = \tilde{g}_{T+1} = 0$.

Now if we combine the above with the left hand side of the result from adding up (19) from $t = 0$ to $T$, teslescoping and dropping $D_\psi(u, x_{T+1}; \phi_{T+1})$ we obtain

$$\sum_{t=1}^{T} \Big( \ell_t(z_t) - \ell_t(u) \Big) + \sum_{t=0}^{T} \langle \tilde{g}_{t+1}, z_t - z_{t+1} \rangle \leq D_\psi(u, z_0; \phi_0) - \sum_{t=0}^{T} D_\psi(z_{t+1}, z_t; \phi_t) + \sum_{t=0}^{T} \langle g_t + \tilde{g}_{t+1} - \tilde{g}_t, z_t - z_{t+1} \rangle$$

$$\overset{\text{①}}{\leq} D_\psi(u, z_0; \phi_0) + \sum_{t=1}^{T} \langle g_t - \tilde{g}_t, z_t - z_{t+1} \rangle,$$

where in ① we drop the Bregman divergence terms and we start the sum from $t = 1$ since it is $0$ for $t = 0$. The inequalities above equal the one in the statement.

$\qquad\square$

**Remark C.5** (Constant step size FTRL as an instance of OMD). *It is well known that FTRL and Mirror Descent are equivalent for constant step sizes in the unconstrained setting, and so are their optimistic variants. With our non-differentiable extension we can see that actually in the constrained*

*case these algorithms are also equivalent, for a specific choice of subgradients in the MD algorithm. Indeed, because of the optimality of $z_t$, we have $0 \in (g_{t-1} + \tilde{g}_t - \tilde{g}_{t-1}) + \partial \psi(z_t) - \phi_{t-1}$, and so $\phi_t \stackrel{\text{def}}{=} \phi_{t-1} - (g_{t-1} + \tilde{g}_t - \tilde{g}_{t-1}) \in \partial \psi(z_t)$ can be our next subgradient. In fact, if we choose $z_0 \in \arg\min_{z \in \mathcal{X}} \psi(z)$, we can make the choice $\phi_0 = 0$, and using the recurrence in the definition of $\phi_t$, we obtain $\phi_t = -\tilde{g}_t - \sum_{i=1}^{t-1} g_i$ by using $g_0 = \tilde{g}_0 = 0$. So the update rule becomes*

$$z_t \in \arg\min_{z \in \mathcal{X}}\{\langle g_{t-1} + \tilde{g}_t - \tilde{g}_{t-1}, z\rangle + \psi(z) - \langle \phi_{t-1}, z\rangle\} = \arg\min_{z \in \mathcal{X}}\left\{\langle \tilde{g}_t + \sum_{i=1}^{t-1} g_i, z\rangle + \psi(z)\right\},$$

*so under this choice the algorithm becomes equivalent to OFTRL with constant regularizer $\psi$. However, note that other choices of subgradients $\phi_t$ yield different update rules.*

## C.2 CONVERGENCE RATE OF OPTIMISTIC FW

Given the theory developed so far in this section, we can now show the convergence of the two variants in our Algorithm 2. We denote $G_t^{\text{OP}}$ the primal-dual gap in (15) when we use either of our two optimistic update rules and where the upper bound of the regret $R_t(x^*)$ is $\widehat{R}_t(x^*) \stackrel{\text{def}}{=} \max_{v \in \mathcal{X}} R_t(v)$.

**Theorem 3.1.** [↓] *Let $\mathcal{X}$ be compact and convex, and let $\psi : \mathcal{X} \to \mathbb{R}$ be a closed convex function, subdifferentiable in $\mathcal{X}$. Let $f$ be convex and $L$-smooth in the set $\mathcal{X}$ of diameter $D$ with respect to a norm $\|\cdot\|$ that we access via an unbiased stochastic gradient oracle $\mathfrak{O}(x)$ with finite variance $\sigma^2 \geq 0$. The iterates $x_t$ of Algorithm 2 satisfy:*

$$\mathbb{E}[f(x_t) - f(x^*)] \leq \mathbb{E}[G_t^{\text{OP}}] \leq \frac{\psi(x^*) - \psi(x_1)}{t(t+1)} + \frac{4LD^2}{t+1} + \frac{\sqrt{2}\sigma D}{2},$$

*for the variant in Line 5. For the variant in Line 6 we obtain the same except that $\psi(x^*) - \psi(x_1)$ is substituted by $D_\psi(x^*, x_0; \phi_0)$, where $\phi_0 \in \partial \psi(x_0)$.*

Generally FW algorithms are applied to optimization problems with compact feasible sets, in which case $x^*$ above exists. However, the generalized FW framework does not assume the feasible set is compact or that $f$ has a minimizer. We note that in the proof, the value of the parameters $a_t$ is not used until the last inequality in the theorem statement. One can also set $a_t = \Theta(t^c)$ for a constant $c > 1$, thus obtaining $A_t = \Theta(t^{c+1})$ and a rate $f(x_t) - f(u) = O(\frac{\psi(u)-\psi(x_1)}{t^{c+1}} + \frac{1}{t^{c+1}}\sum_{i=1}^{t}\frac{LD^2 i^{2c}}{i^{c+1}}) = O(\frac{\psi(u)-\psi(x_1)}{t^{c+1}} + \frac{LD^2}{t})$. Thus, we can reduce the influence of $\psi$ on the convergence rate polynomially.

We also note that the most common use of Frank-Wolfe corresponds to $\psi$ being the indicator function of a convex compact set $\mathcal{X}$, in which case we obtain $f(x_t) - f(x^*) \leq \frac{4LD^2}{t+1}$.

*Proof.* (Theorem 3.1) We start proving the statement for the choice of OFTRL in Line 5 of Algorithm 2 that prescribes how the points $v_i$ are defined. Denoting $R_t(u)$ the corresponding regret at $u$ after $t$ steps, taking into account that we use $a_i g_i$ as hints for $i \geq 1$, satisfying $\mathbb{E}[g_i] = \nabla f(x_{i-1})$, defining $\tilde{v}_{i+1} = v_{i+1}$ for $i < t$ and $\tilde{v}_{t+1} = u$, and using $\xi_i \stackrel{\text{def}}{=} g_{i+1} - \nabla f(x_i)$ for the noise term, we

obtain

$$
\begin{aligned}
\mathbb{E}[f(x_t) - f(u)] \leq \mathbb{E}[G_t^{\mathrm{OP}}] &\overset{\text{①}}{\leq} \mathbb{E}\left[\frac{R_t(u)}{A_t}\right] \\
&\overset{\text{②}}{\leq} \frac{\psi(u) - \psi(x_1)}{A_t} + \frac{1}{A_t}\sum_{i=1}^{t}\left(a_i\langle\nabla f(x_i) - \nabla f(x_{i-1}), v_i - \tilde{v}_{i+1}\rangle + \mathbb{E}[\langle\xi_i - \xi_{i-1}, v_i - \tilde{v}_{i+1}\rangle]\right) \\
&\overset{\text{③}}{\leq} \frac{\psi(u) - \psi(x_1)}{A_t} + \frac{1}{A_t}\sum_{i=1}^{t}\left(a_i L\|x_i - x_{i-1}\|D + \mathbb{E}\left[\frac{a_i\eta}{2}\|\xi_i - \xi_{i-1}\|_*^2\right] + \frac{a_i D^2}{2\eta}\right) \\
&\overset{\text{④}}{\leq} \frac{\psi(u) - \psi(x_1)}{A_t} + \frac{1}{A_t}\sum_{i=1}^{t}\left(\frac{a_i^2 LD^2}{A_i} + a_i\eta\sigma^2 + \frac{a_i D}{2\eta}\right) \\
&\overset{\text{⑤}}{\leq} \frac{\psi(u) - \psi(x_1)}{t(t+1)} + \frac{1}{t(t+1)}\sum_{i=1}^{t}\left(\frac{4iLD^2}{i+1} + \frac{\sqrt{2}a_i\sigma D}{2}\right) \\
&\overset{\text{⑥}}{<} \frac{\psi(u) - \psi(x_1)}{t(t+1)} + \frac{4LD^2}{t+1} + \frac{\sqrt{2}\sigma D}{2}.
\end{aligned}
\tag{20}
$$

where ① uses (15) which holds in expectation for stochastic our gradients $g_i$, taking into account that our algorithm has the form described at the beginning of Appendix C for the points $v_t$ running the OFTRL algorithm. Inequality ② adds and subtracts terms to isolate the random part, and also uses Corollary C.2, since our points $v_i$ are computed according to the OFTRL algorithm. Then, ③ uses Hölder's inequality, $L$-Lipschitzness of $\nabla f(\cdot)$, and bounds $\|v_i - \tilde{v}_{i+1}\| \leq D$ for all $i \in [t]$. Note the inequality holds for all $\eta > 0$. Then, inequality ④ uses that by definition of $x_i$, we have $x_i - x_{i-1} = \frac{a_i}{A_i}(v_i - x_{i-1})$ for $i \geq 1$, bounds $\|v_i - x_{i-1}\| \leq D$ and uses the triangular inequality, $(a+b)^2 \leq 2a^2 + 2b^2$, and the bounded variance assumption. This yields the first part of the theorem statement. Now, substituting the choices of $a_t = 2t$, and thus $A_t = \sum_{i=0}^{t} a_i = t(t+1)$ we obtain ⑤ by choosing the best $\eta > 0$ that optimizes the bound, and a computation gives ⑥.

The proof for the OMD variant in Line 6 of Algorithm 2 is identical, except that in ② above we use Theorem C.4 and so $\tilde{v}_{t+1}$ changes to be a point in $\arg\min_{v\in\mathbb{R}^d}\{\langle g_t - \tilde{g}_t, v\rangle + D_\psi(z, v_t; \phi_t)\}\}$, that is, it corresponds to the next point computed with the update rule when we choose $\tilde{g}_{t+1} = 0$. And also, we have $D_\psi(u, x_0)$ instead of $\psi(u) - \psi(x_1)$. The rest of the inequalities in (20) are the same. $\qquad\square$

**Remark C.6** (Computable primal-dual gap). *Note that for $\psi$ being the indicator of a compact set, and for $u = x^* \in \arg\min_{x\in\mathrm{dom}(\psi)} f(x)$, we have for the OFTRL variant the upper bound on the primal-dual gap $\frac{1}{A_t}\sum_{i=1}^{t} a_i\langle\nabla f(x_i) - g_i, v_i - \tilde{v}_{i+1}\rangle - A_{i-1}D_f(x_{i-1}, x_i)$, which depends on the unknown point $x^*$, since $\tilde{v}_{t+1} = x^*$. For a computable primal-dual gap, we can apply an analogous bound to ③ in (20) but for the last summand only, that is*

$$
G_t \leq \frac{1}{A_t}\left(\sum_{i=1}^{t-1} a_i\langle\nabla f(x_i) - g_i, v_i - v_{i+1}\rangle + \|\nabla f(x_t) - \nabla f(x_{t-1})\|_* D - \sum_{i=1}^{t} A_{i-1}D_f(x_{i-1}, x_i)\right).
$$

*Alternatively, using OMD and taking one more linear minimization oracle to compute $\tilde{v}_{t+1}$, we already have that our bound is a computable primal-dual gap: $\frac{1}{A_t}\sum_{i=1}^{t} a_i\langle\nabla f(x_i) - g_i, v_i - \tilde{v}_{i+1}\rangle - A_{i-1}D_f(x_{i-1}, x_i)$. We note that it is also possible to obtain an analogous different $\tilde{v}_{t+1}$ for FTRL that does not depend on $x^*$, at the expense of computing a linear minimization oracle. Just take the equivalence of OFTRL and OMD for specific choices of the subgradients $\phi_t$ in Remark C.5.*

## D    PROOFS FOR PRIMAL-DUAL SHORT-STEPS

*Proof.* (Proposition 4.1) In (8) or (11), after their respective inequalities ①, isolating $G_t$, using $\gamma_t \stackrel{\text{def}}{=} \frac{a_t}{A_t}$, which ranges from $[0, 1)$ as $a_t \in [0, \infty)$, we obtain

$$G_t \leq \frac{A_{t-1}G_{t-1}}{A_t} - \frac{a_t}{A_t}\langle \nabla f(x_t), v_t - x_t \rangle + (f(x_{t+1}) - f(x_t))$$
$$= (1 - \gamma_t)G_{t-1} - \gamma_t\langle \nabla f(x_t), v_t - x_t \rangle + f((1 - \gamma_t)x_t + \gamma_t v_t) - f(x_t).$$

Differentiating the right hand side twice with respect to $\gamma_t$, we obtain

$$\langle \nabla^2 f((1 - \gamma_t)x_t + \gamma_t v_t)(v_t - x_t), v_t - x_t \rangle \geq 0,$$

hence the expression is convex with respect to $\gamma_t$, which proves the first statement.

For the second one, we already performed some steps of the proof in the main paper. Using the primal-dual analysis of FW and HB-FW Appendix B and a few computations we arrive to (4). Plugging the choice of our primal-dual short step (5) into (4) and simplifying leads to

$$G_t \leq \left(1 - \frac{\gamma_t}{2}\right)G_{t-1} = \left(1 - \frac{\min\{1, \frac{G_{t-1}}{L\|v_t - x_t\|^2}\}}{2}\right)G_{t-1},$$

or equivalently

$$G_{t-1} - G_t \geq \frac{1}{2}G_{t-1}\min\{1, \frac{G_{t-1}}{L\|v_t - x_t\|^2}\}.$$

Now, one can apply (Braun et al., 2022, Lemma 2.21) (or similar results; see (Garber & Hazan, 2015)), which converts the contraction inequality into a convergence guarantee, so we obtain:

$$G_t \leq \frac{4LD^2}{t + 2},$$

for the primal-dual gap convergence rate after bounding $\|v_i - x_i\|^2 \leq D^2$ for all $i \leq t$.

The progress made by the line search in terms of primal-dual gap is greater than the guaranteed progress (4) used by this second approach and so the line search variant also enjoys the same rates of convergence. □

**Remark D.1** (Using $f(x^*)$ for the gap). *The primal-dual short step is a generalization of the standard short steps, since if we choose the best possible lower bound $L_t \stackrel{\text{def}}{=} f(x^*)$, which is a value that we do not know in general, and if we define the gap accordingly $G_t \stackrel{\text{def}}{=} f(x_{t+1}) - f(x^*)$, then we obtain*

$$A_t G_t - A_{t-1}G_{t-1} = A_t f(x_{t+1}) - A_{t-1}f(x_t) - a_t f(x^*),$$

*which after using smoothness and reorganizing yields*

$$G_t \leq (1 - \gamma_t)G_{t-1} + \gamma_t(f(x_t) - f(x^*)) + \gamma_t\langle \nabla f(x_t), v_t - x_t \rangle + \gamma_t^2\frac{L}{2}\|v_t - x_t\|^2$$

$$\stackrel{①}{=} G_{t-1} + \gamma_t\langle \nabla f(x_t), v_t - x_t \rangle + \gamma_t^2\frac{L}{2}\|v_t - x_t\|^2 \tag{21}$$

*where ① holds by definition of the gap $G_{t-1} = f(x_t) - f(x^*)$. Note that optimizing the right hand side of the last expression results into regular short steps, which can be computed even if we do not know the value of $L_t \stackrel{\text{def}}{=} f(x^*)$. This computation is simply saying the natural fact that if our primal-dual gap becomes the primal gap, then these new short steps that greedily maximize guaranteed progress on $G_t$, become the regular short steps, that greedily maximize primal progress.*

### D.1    DETAILS ON THE PRIMAL-DUAL STEP SIZE FOR GRADIENT DESCENT

First recall a few properties of the classical gradient descent (GD) algorithm, with arbitrary step sizes $a_t$:

$$x_{t+1} \stackrel{\text{def}}{=} x_t - a_t\nabla f(x_t) = \arg\min_{x \in \mathbb{R}^d}\left\{a_t\langle \nabla f(x_t), x - x_t \rangle + \frac{1}{2}\|x - x_t\|_2^2\right\}$$

$$= \arg\min_{x \in \mathbb{R}^d}\left\{\sum_{i=1}^{t}a_i\langle \nabla f(x_i), x - x_i \rangle + \frac{1}{2}\|x - x_1\|_2^2\right\} = x_0 - \sum_{i=1}^{t}a_i\nabla f(x_i).$$

*Proof.* ([Proposition 4.2])

Recall our notation of positive weights $a_t$ and $A_t \overset{\text{def}}{=} \sum_{i=1}^{t} a_i$. The lower bound that we use to define the primal-dual gap is obtained from

$$
\begin{aligned}
A_t f(x^*) &\overset{\text{①}}{\geq} \sum_{i=1}^{t} a_i f(x_i) + \sum_{i=1}^{t} a_i \langle \nabla f(x_i), x^* - x_i \rangle \pm \frac{1}{2}\|x^* - x_1\|_2^2 \\
&\overset{\text{②}}{\geq} \sum_{i=1}^{t} a_i f(x_i) + \min_{x \in \mathbb{R}^d} \left\{ \sum_{i=1}^{t} a_i \langle \nabla f(x_i), x - x_i \rangle + \frac{1}{2}\|x - x_1\|_2^2 \right\} - \frac{1}{2}D^2 \\
&\overset{\text{③}}{=} \sum_{i=1}^{t} a_i f(x_i) + \sum_{i=1}^{t} a_i \langle \nabla f(x_i), x_{t+1} - x_i \rangle + \frac{1}{2}\|x_{t+1} - x_1\|_2^2 - \frac{1}{2}D^2 \overset{\text{def}}{=} A_t L_t.
\end{aligned}
$$

Where ① is due to convexity, and in ② and ③ we take a minimum and use the definition of $x_{t+1}$ as a minimizer of $\Lambda_t(x) \overset{\text{def}}{=} \sum_{i=1}^{t} a_i \langle \nabla f(x_i), x - x_i \rangle + \frac{1}{2}\|x - x_1\|_2^2$. We also used the bound $D \geq \|x^* - x_1\|_2$. Now define the gap $G_t \overset{\text{def}}{=} f(x_{t+1}) - L_t$. We have

$$
\begin{aligned}
A_t G_t - A_{t-1} G_{t-1} - 1_{\{t=1\}} \frac{1}{2}D^2 &\overset{\text{①}}{=} A_t(f(x_{t+1}) - f(x_t)) + \cancel{a_t f(x_t)} \\
&\quad - \left( \cancel{\sum_{i=1}^{t} a_i f(x_i)} + \sum_{i=1}^{t-1} a_i \langle \nabla f(x_i), x_{t+1} - x_i \rangle + \frac{1}{2}\|x_{t+1} - x_1\|_2^2 \right) - a_t \langle \nabla f(x_t), x_{t+1} - x_t \rangle \\
&\quad + \left( \cancel{\sum_{i=1}^{t-1} a_i f(x_i)} + \sum_{i=1}^{t-1} a_i \langle \nabla f(x_i), x_t - x_i \rangle + \frac{1}{2}\|x_t - x_1\|_2^2 \right) \\
&\overset{\text{②}}{\leq} \langle \nabla f(x_t), A_t(x_{t+1} - x_t) - a_t(x_{t+1} - x_t) \rangle + \frac{A_t L}{2}\|x_{t+1} - x_t\|_2^2 - \frac{1}{2}\|x_{t+1} - x_t\|_2^2 \\
&\overset{\text{③}}{=} \|\nabla f(x_t)\|_2^2 \left( -a_t A_{t-1} + \frac{a_t^2 A_t L}{2} - \frac{a_t^2}{2} \right) \overset{\text{def}}{=} E_t(a_t).
\end{aligned}
\tag{22}
$$

Above, in ① we write out the definitions and cancel some terms. In ②, we used smoothness for the first term, and we used the 1-strong convexity of $\Lambda_{t-1}(x)$ and the fact that $x_t$ is its minimizer so $\Lambda_{t-1}(x_t) - \Lambda_{t-1}(x_{t+1}) \leq -\frac{1}{2}\|x_{t+1} - x_t\|_2^2$. In ③, we used the definition of $x_{t+1}$ and grouped terms. This time, we have defined the error $E_t(a_t)$ as a function of $a_t$.

We can apply the same technique as in the primal dual steps for Frank-Wolfe algorithms. Let $\mathcal{G}_{t-1} \overset{\text{def}}{=} A_{t-1} G_{t-1} + 1_{\{t=1\}} \frac{1}{2}D^2 \geq A_{t-1}(f(x_{t+1}) - f(x^*)) \geq 0$. Hence, for $t \geq 1$, we aim to minimize the right hand side of the following that comes reorganizing the above

$$
G_t \leq \frac{\mathcal{G}_{t-1}}{A_t} + \|\nabla f(x_t)\|_2^2 \left( -\frac{a_t A_{t-1}}{A_t} + \frac{a_t^2 L}{2} - \frac{a_t^2}{2A_t} \right).
\tag{23}
$$

Differentiating the RHS with respect to $a_t$ and equating to 0 (recall $A_t = A_{t-1} + a_t$) we obtain

$$
\begin{aligned}
0 &= -\frac{\mathcal{G}_{t-1}}{A_t^2} + \|\nabla f(x_t)\|_2^2 \left( -\frac{A_{t-1} A_t - a_t A_{t-1}}{A_t^2} + a_t L - \frac{2a_t A_t - a_t^2}{2A_t^2} \right) \\
&\iff 0 = -2\mathcal{G}_{t-1} + \|\nabla f(x_t)\|_2^2 \left( -2A_{t-1}^2 + 2a_t A_t^2 L - 2a_t A_t + a_t^2 \right),
\end{aligned}
\tag{24}
$$

which gives a cubic equation for $a_t$. From now on, assume $a_t$ has the value of this minimizer. Note that this solution is always greater than $\frac{1}{2L}$, for all $t \geq 1$. Indeed, first notice that the right hand side of (23) is convex on $a_t$ for $a_t > 0$, since if we differentiate the right hand side a second time we obtain:

$$
2\frac{\mathcal{G}_{t-1}}{A_t^3} + \|\nabla f(x_t)\|_2^2 \left( \frac{2A_{t-1}^2}{A_t^3} + L - \frac{A_{t-1}^2}{A_t^3} \right) \geq 0.
$$

Given this convexity, in order to show that the solution of (24) is $a_t > \frac{1}{2L}$, it is enough to check that the derivative is negative at $a_t = \frac{1}{2L}$ which is immediate after substitution. This fact yields the convergence rate:

$$f(x_{t+1}) - f(x_t) \leq G_t \overset{①}{\leq} \frac{\mathcal{G}_{t-1}}{A_t} + \|\nabla f(x_t)\|_2^2 \left( -\frac{a_t A_{t-1}}{A_t} + \frac{a_t^2 L}{2} - \frac{a_t^2}{2A_t} \right)$$

$$\overset{②}{\leq} \frac{\mathcal{G}_{t-1}}{A_{t-1} + 1/(2L)} \overset{③}{\leq} \frac{\mathcal{G}_0}{t/(2L)}$$

$$\overset{④}{=} \frac{LD^2}{t},$$

where ① is (23), ② holds by substituting the minimizer $a_t$ by $1/(2L)$ and droping $E_t(1/(2L))$ which after substitution it is immediate to see that it is nonpositive. Inequality ③ holds by applying recursively ① and ②, taking into account that $\mathcal{G}_k = A_k G_k$ for all $k > 1$. And finally ④ substitutes the value of $\mathcal{G}_0$.

**Line search for minimizing the primal-dual gap**  Recall that we define $\mathcal{G}_{t-1} \overset{\text{def}}{=} A_{t-1} G_{t-1} + 1_{\{t=1\}} \frac{1}{2} D^2 \geq 0$. If for $t \geq 1$, after ① in (22) we do not apply smoothness but only apply the inequality $\Lambda_{t-1}(x_t) - \Lambda_{t-1}(x_{t+1}) \leq -\frac{1}{2}\|x_{t+1} - x_t\|^2$, use $x_{t+1} - x_t = -a_t \nabla f(x_t)$ and then isolate $G_t$, we obtain

$$G_t \leq \frac{\mathcal{G}_{t-1}}{A_t} + \frac{a_t^2}{2A_t}\|\nabla f(x_t)\|^2 + (f(x_{t+1}) - f(x_t)) \tag{25}$$

If we differentiate twice with respect to $a_t$, we obtain (taking into account $A_t$ and $x_{t+1}$ depend on $a_t$):

$$2\frac{\mathcal{G}_{t-1}}{A_t^3} + \frac{A_{t-1}^2}{A_t^3}\|\nabla f(x_t)\|^2 + \langle \nabla^2 f(x_{t+1})\nabla f(x_t), \nabla f(x_t) \rangle \geq 0.$$

Thus, the right hand side of (25) is convex with respect to the variable $a_t$, which means that we can do a line search in order to greedily minimize the primal-dual gap progress at each iteration. Above, we used twice differentiability of $f$ for the last summand but in fact this is not required, since $f(x_{t+1}) = f(x_t - a_t \nabla f(x_t))$ is clearly convex on $a_t$ due to the convexity of $f$ restricted to the line $a_t \mapsto x_t - a_t \nabla f(x_t)$.

Note that the upper bound in (25) is tighter than the one in (23), and thus, the convergence rate from the line search on the primal-dual gap bound is at least the one from the primal-dual short step that minimizes (23).  □

# E  EXTENDING PRIMAL ANALYSES OF KNOWN FW CONVERGENCE RESULTS TO PRIMAL-DUAL ANALYSES

In this section we demonstrate via some examples how primal-dual analyses can be derived from existing primal analyses. We picked examples that demonstrate relevant aspects of the argument, so that the interested reader should be able to carry them over to other settings. We also refer the interested reader to Braun et al. (2022) for full details on the primal convergence analyses.

In line with Section 2, we let $g(x_\ell) = \max_{v \in \mathcal{X}} \langle \nabla f(x_\ell), x_\ell - v \rangle = \langle \nabla f(x_\ell), x_\ell - v_\ell \rangle$ be the FW gap and

$$A_\ell L_\ell = \sum_{i=0}^{\ell} a_i f(x_i) + \sum_{i=0}^{\ell} a_i \langle \nabla f(x_i), v_i - x_i \rangle,$$

be the associated lower bound function for the standard FW gap. Together with $G_\ell = f(x_{\ell+1}) - L_\ell$ we can now rewrite the crucial term $A_\ell G_\ell - A_{\ell-1} G_{\ell-1} \leq E_\ell$ appearing in primal-dual analyses as follows:

$$A_\ell(f(x_{\ell+1}) - L_\ell) - A_{\ell-1}(f(x_\ell) - L_{\ell-1})$$
$$= A_\ell f(x_{\ell+1}) - A_{\ell-1}f(x_\ell) - (A_\ell L_\ell - A_{\ell-1}L_{\ell-1})$$
$$= A_\ell(f(x_{\ell+1}) - f(x_\ell)) + a_\ell g(x_\ell) \leq E_\ell.$$

This can be rearranged to the following fundamental bound inequality:

$$\underbrace{a_\ell g(x_\ell)}_{\text{weighted gap}} - \underbrace{A_\ell(f(x_\ell) - f(x_{\ell+1}))}_{\text{weighted progress}} \leq E_\ell. \tag{26}$$

From this inequality there are various ways to proceed. The most natural way is often to plug-in the progress guarantee from the short-step for a direction $d_t$ (typically $d_t = x_t - v_t$ but not always), that is, we use smoothness and optimize the bound:

$$f(x_{\ell+1}) - f(x_\ell) \leq -\frac{\langle \nabla f(x_\ell), d_\ell \rangle^2}{2L\|d_\ell\|^2}.$$

Chaining this bound into (26) makes the left-hand side only larger, so that we obtain:

$$\underbrace{a_\ell g(x_\ell)}_{\text{weighted gap}} - A_\ell \underbrace{\frac{\langle \nabla f(x_\ell), d_\ell \rangle^2}{2L\|d_\ell\|^2}}_{\text{weighted progress}} \leq E_\ell. \tag{27}$$

Inequalities (26) and (27) will be key in the following to simply transfer primal convergence results into primal-dual convergence results. To this end the following lemma will be useful, which turns the relation between $a_\ell$ and $A_\ell$ into a convergence rate.

**Lemma E.1** (Linear rate conversion). *Suppose that $a_\ell(\kappa - 1) = A_{\ell-1}$ (or equivalently: $a_\ell \kappa = A_\ell$) then it holds*

$$1/A_\ell = \left(1 - \frac{1}{\kappa}\right) 1/A_{\ell-1}.$$

*Proof.* The proof is by straightforward rearranging:

$$a_\ell(\kappa - 1) = A_{\ell-1}$$
$$\Rightarrow A_\ell(\kappa - 1) - A_{\ell-1}(\kappa - 1) = A_{\ell-1}$$
$$\Rightarrow A_\ell(\kappa - 1) = A_{\ell-1}\kappa$$
$$\Rightarrow A_\ell = \frac{\kappa}{\kappa - 1}A_{\ell-1}$$
$$\Rightarrow 1/A_\ell = \left(1 - \frac{1}{\kappa}\right) 1/A_{\ell-1}$$

$\square$

### E.1 LINEAR CONVERGENCE RATES: THE CASE $E_\ell = 0$

We will first consider the case where we can set $E_\ell = 0$. This case, which usually comes with a simpler analysis basically captures all linear convergence rate results for FW algorithms. To this end we present a generic argument that then allows us to simply reuse already known analyses for primal convergence to establish primal-dual convergence simply by plugging in the bounds on the primal progress and the dual gap; it is important to note that these are usually known already from the primal analysis. We start from (26):

$$a_\ell g(x_\ell) - A_\ell(f(x_\ell) - f(x_{\ell+1})) \leq E_\ell,$$

and we set $E_\ell = 0$ and solve for equality. For the sake of exposition here and in the following let $p_\ell \stackrel{\text{def}}{=} f(x_\ell) - f(x_{\ell+1})$ denote the *primal progress* in iteration $\ell$. We obtain:

$$a_\ell g(x_\ell) - A_\ell p_\ell = 0 \tag{28}$$
$$\Rightarrow a_\ell g(x_\ell) = A_\ell p_\ell \tag{29}$$
$$\Rightarrow a_\ell (g(x_\ell) - p_\ell) = A_{\ell-1} p_\ell \tag{30}$$
$$\Rightarrow a_\ell \left(\frac{g(x_\ell)}{p_\ell} - 1\right) = A_{\ell-1}, \tag{31}$$

and combining this with Lemma E.1, we obtain:

**Lemma E.2** (Transfer template for bounds). *Let $g(x)$ be a dual bound and let $p_\ell$ denote the primal progress in iteration $\ell$. If we set $E_\ell = 0$, then we obtain:*

$$1/A_\ell = \left(1 - \frac{p_\ell}{g(x_\ell)}\right) 1/A_{\ell-1}.$$

Also observe that we can use (31) to dynamically compute the $a_\ell$ from an actual step which allows us to report an adaptive gap if desired.

### E.1.1 $f$ STRONGLY CONVEX AND $x^* \in \mathrm{rel.\,int}(P)$

For the sake of completeness, we will present two ways of deriving primal-dual convergence rates for the case where $f$ is strongly convex and $x^* \in \mathrm{rel.\,int}(P)$. First, we will go the "complicated" route, without relying on Lemma E.2 and instead directly solving (27) for $a_\ell$. Then, we will use Lemma E.2 to obtain the (same) convergence rate. The primal analysis is due to Guélat & Marcotte (1986).

We assume that $f$ is $\mu$-strongly convex and $B(x^*, r) \subseteq \mathcal{X}$. We start from the primal gap bound via strong convexity:

$$f(x) - f(x^*) \leq \frac{\langle \nabla f(x), x - x^* \rangle^2}{2\mu \|x - x^*\|^2},$$

where $\mu > 0$ is the strong convexity constant, which we then combine with the *scaling inequality* for $x^* \in \mathrm{rel.\,int}(P)$ from Guélat & Marcotte (1986):

$$\frac{r}{D} \frac{\langle \nabla f(x), x - x^* \rangle}{\|x - x^*\|} \leq \frac{\langle \nabla f(x), x - v \rangle}{\|x - v\|},$$

where $v$ is the FW-vertex for $x$, $D$ is the diameter, and $r$ is radius of the ball around the optimal solution $x^*$ contained in $P$, to obtain the bound on the primal gap:

$$f(x) - f(x^*) \leq \frac{D^2}{r^2} \frac{\langle \nabla f(x), x - v \rangle^2}{2\mu \|x - v\|^2},$$

and we set the gap function $g(x_\ell)$ to be the right hand side of the above inequality:

$$g(x_\ell) \stackrel{\mathrm{def}}{=} \frac{D^2}{r^2} \frac{\langle \nabla f(x_\ell), x_\ell - v_\ell \rangle^2}{2\mu \|x_\ell - v_\ell\|^2},$$

where $x_\ell, v_\ell$ are the iterate and FW-vertex in the $\ell$-th iteration, respectively. Now for the direction $d_\ell$ we pick the standard FW direction, i.e., $d_\ell = x_\ell - v_\ell$. With this choice (27) becomes

$$a_\ell g(x_\ell) - A_\ell \frac{\langle \nabla f(x_\ell), d_\ell \rangle^2}{2L \|d_\ell\|^2} \leq E_\ell$$

$$\Rightarrow a_\ell \frac{D^2}{r^2} \frac{\langle \nabla f(x_\ell), x_\ell - v_\ell \rangle^2}{2\mu \|x_\ell - v_\ell\|^2} - A_\ell \frac{\langle \nabla f(x_\ell), x_\ell - v_\ell \rangle^2}{2L \|x_\ell - v_\ell\|^2} \leq E_\ell$$

$$\Rightarrow \frac{\langle \nabla f(x_\ell), x_\ell - v_\ell \rangle^2}{\|x_\ell - v_\ell\|^2} \left( a_\ell \frac{1}{2\mu} \frac{D^2}{r^2} - A_\ell \frac{1}{2L} \right) \leq E_\ell.$$

We want $a_\ell \frac{1}{2\mu} \frac{D^2}{r^2} - A_\ell \frac{1}{2L} = 0$, and in particular then we can choose $E_\ell = 0$. We obtain:

$$a_\ell \frac{1}{2\mu} \frac{D^2}{r^2} - A_\ell \frac{1}{2L} = 0$$

$$\Rightarrow a_\ell \frac{1}{2\mu} \frac{D^2}{r^2} = A_\ell \frac{1}{2L}$$

$$\Rightarrow a_\ell \left( \frac{1}{2\mu} \frac{D^2}{r^2} - \frac{1}{2L} \right) = A_{\ell-1} \frac{1}{2L}$$

$$\Rightarrow a_\ell \left( \frac{L}{\mu} \frac{D^2}{r^2} - 1 \right) = A_{\ell-1},$$

and applying Lemma E.1, we obtain immediately that

$$1/A_\ell = \left(1 - \frac{\mu}{L}\frac{r^2}{D^2}\right)1/A_{\ell-1},$$

and hence the expected linear rate of convergence follows for the primal-dual gap $G_t$, i.e.,

$$G_t \le \frac{A_{t-1}}{A_t}G_{t-1} = \left(1 - \frac{\mu}{L}\frac{r^2}{D^2}\right)G_{t-1}.$$

Alternatively, we could have directly applied Lemma E.2 to

$$p_\ell = \frac{\langle\nabla f(x_\ell), x_\ell - v_\ell\rangle^2}{2L\|x_\ell - v_\ell\|^2} \qquad \text{and} \qquad g(x_\ell) = \frac{D^2}{r^2}\frac{\langle\nabla f(x_\ell), x_\ell - v_\ell\rangle^2}{2\mu\|x_\ell - v_\ell\|^2},$$

to obtain the same result.

### E.1.2   AWAY-STEP VARIANTS OVER POLYTOPES AND STRONGLY CONVEX FUNCTIONS

Let $P$ be a polytope and $f$ be $\mu$-strongly convex. The proof is similar to Appendix E.1.1. We pick

$$g(x_\ell) \stackrel{\text{def}}{=} \frac{\langle\nabla f(x_\ell), s_\ell - v_\ell\rangle^2}{2\mu w(P)^2},$$

which is the *geometric strong convexity* that arises from combining the strong convexity bound in Appendix E.1.1 with a different scaling inequality, namely the one that appears in active set based, away-step inducing variants:

$$\langle\nabla f(x_\ell), s_\ell - v_\ell\rangle \ge w(P)\frac{\langle\nabla f(x_\ell), x_\ell - x^*\rangle}{\|x_\ell - x^*\|},$$

where $w(P)$ is the *pyramidal width* of $P$. Moreover, we pick $d_\ell = s_\ell - v_\ell$ and use a slightly different bound on the primal progress, namely

$$p_\ell = \frac{\langle\nabla f(x_\ell), s_\ell - v_\ell\rangle^2}{4L\|s_\ell - v_\ell\|^2},$$

note the extra factor of 2 in the denominator, which is due to the selection between the away-step or the FW step; we refer the reader to Lacoste-Julien & Jaggi (2015) and Braun et al. (2022) for more details on both the geometric strong convexity as well as the modified primal progress bound; see also Peña & Rodríguez (2018) for a unified perspective on the different notions of geometric conditioning arising from the polytope. We also ignore drop steps here for the sake of simplicity, however they can be added easily and deteriorite the number of required steps only by a constant factor of 2; see Braun et al. (2022) for more details.

We apply Lemma E.2 to

$$p_\ell = \frac{\langle\nabla f(x_\ell), s_\ell - v_\ell\rangle^2}{4L\|s_\ell - v_\ell\|^2} \qquad \text{and} \qquad g(x_\ell) = \frac{\langle\nabla f(x_\ell), s_\ell - v_\ell\rangle^2}{2\mu w(P)^2},$$

to obtain

$$1/A_\ell = \left(1 - \frac{\mu}{L}\frac{w(P)^2}{2D^2}\right)1/A_{\ell-1},$$

as expected but again for the primal-dual gap.

### E.1.3   STRONGLY CONVEX FEASIBLE REGION AND $\|\nabla f(x)\| \ge c > 0$

We now consider the case where the feasible region is strongly convex and the unconstrained minimizer of $f$ lies outside of the feasible region. Since the feasible region is strongly convex we obtain the scaling inequality (Garber & Hazan, 2015; Braun et al., 2022):

$$\frac{\alpha}{4}\|\nabla f(x_t)\| \le \frac{\langle\nabla f(x_t), x_t - v_t\rangle}{\|x_t - v_t\|^2}.$$

Moreover we have

$$f(x_t) - f(x^*) \leq \langle \nabla f(x_t), x_t - v_t \rangle.$$

Combining the two leads to:

$$(f(x_t) - f(x^*)) \frac{\alpha}{4} \|\nabla f(x_t)\| \leq \frac{\langle \nabla f(x_t), x_t - v_t \rangle^2}{\|x_t - v_t\|^2},$$

which gives the dual gap bound:

$$f(x_t) - f(x^*) \leq \frac{4}{\alpha \|\nabla f(x_t)\|} \frac{\langle \nabla f(x_t), x_t - v_t \rangle^2}{\|x_t - v_t\|^2}, \tag{32}$$

and we set

$$g(x_t) \stackrel{\text{def}}{=} \frac{4}{\alpha \|\nabla f(x_t)\|} \frac{\langle \nabla f(x_t), x_t - v_t \rangle^2}{\|x_t - v_t\|^2}.$$

Moreover, for the primal progress bound we simply pick the short-step with the standard direction $x_t - v_t$, i.e.,

$$p_t \stackrel{\text{def}}{=} \frac{\langle \nabla f(x_\ell), x_\ell - v_\ell \rangle^2}{2L \|x_\ell - v_\ell\|^2}.$$

Applying Lemma E.2 we immediately obtain:

$$1/A_\ell = \left( 1 - \frac{\alpha \|\nabla f(x_t)\|}{8L} \right) 1/A_{\ell-1},$$

and using $\|\nabla f(x)\| \geq c > 0$ we have

$$1/A_\ell \leq \left( 1 - \frac{\alpha c}{8L} \right) 1/A_{\ell-1},$$

which completes the argument.

## E.2 VARIANTS UTILIZING BREGMAN DIVERGENCES: THE CASE WHERE $E_\ell \neq 0$

We will now consider the slightly more involved case where $E_\ell \neq 0$. This means that we cannot simply apply Lemma E.2 but rather have to account for the error term $E_\ell$. To this end, we first collect alternative choices of $a_t$, $A_t$, and $\gamma_t$, assuming linear coupling $\gamma_t = a_t/A_t$, that will become useful in the rest of the section.

**Lemma E.3** (Open-loop step-sizes). *Let $1 \leq \ell \in \mathbb{N}$ and define $\gamma_t = \frac{\ell}{t+\ell}$. Then*

$$a_t = a_0 \binom{t + \ell - 1}{t} \quad and \quad A_t = a_0 \binom{t + \ell}{t},$$

*is a valid choice and moreover it holds:*

1. $A_t = a_t \frac{t+\ell}{\ell}$;

2. $\frac{a_t^2}{A_t} = \gamma_t a_t = \frac{\ell}{t+\ell} a_t$;

3. *With the choice $a_0 = \ell!$, we recover for $\ell = 2$ the known values $a_t = 2(t + 2)$ and $A_t = (t + 1)(t + 2)$.*

*Proof.* Verified by straightforward calculation. $\qquad \square$

The next lemma is folklore and we include it for completeness.

**Lemma E.4** (Power Sum Estimations). *The following holds:*

1.

$$\sum_{t=0}^{T} t^\alpha \begin{cases} \approx \ln(T) + \kappa \leq \ln(T) + 1 & for \; \alpha = -1 \\ = T + 1 & for \; \alpha = 0 \\ \approx \frac{T^{\alpha+1}}{\alpha+1} + \frac{T^\alpha}{2} + O(T^{\alpha-1}) & for \; \alpha > 0 \end{cases},$$

*where $\kappa \approx 0.577$ is the Euler-Mascheroni constant.*

2.

$$\int_0^T x^\alpha dx = \frac{T^{\alpha+1}}{\alpha+1}$$

3.

$$\sum_{t=0}^T t^\alpha \le \int_0^{T+1} x^\alpha dx = \frac{(T+1)^{\alpha+1}}{\alpha+1}$$

*Proof.* The approximation is via Faulhaber's formula. □

As an example, in this section we specifically consider convergence rates arising from the so-called strong growth property as studied in Peña (2023) and Wirth et al. (2024) for FW algorithms. Our starting point is the following simple Bregman expansion of $f$:

$$D_f(y, x) = f(y) - f(x) - \langle \nabla f(x), y - x \rangle.$$

We use the standard FW gap (or alternatively the Heavy-Ball variant) as a lower bound as before, so that the critical estimation becomes, with the usual $x_{t+1} = (1 - \gamma_t)x_t + \gamma_t v_t$:

$$A_t(f(x_{t+1}) - f(x_t)) - a_t\langle \nabla f(x_t), v_t - x_t \rangle$$
$$= A_t(\langle \nabla f(x_t), x_{t+1} - x_t \rangle + D_f(x_{t+1}, x_t)) - a_t\langle \nabla f(x_t), v_t - x_t \rangle.$$

Instead of using the standard linear coupling $\gamma_t = a_t/A_t$, we use the modified linear coupling $\gamma_t = 2a_t/A_t$, so that we obtain:

$$A_t(\langle \nabla f(x_t), x_{t+1} - x_t \rangle + D_f(x_{t+1}, x_t)) - a_t\langle \nabla f(x_t), v_t - x_t \rangle \tag{33}$$
$$= A_t D_f(x_{t+1}, x_t) + 2a_t\langle \nabla f(x_t), v_t - x_t \rangle - a_t\langle \nabla f(x_t), v_t - x_t \rangle \tag{34}$$
$$= A_t D_f(x_{t+1}, x_t) - a_t\langle \nabla f(x_t), x_t - v_t \rangle. \tag{35}$$

The $(M, r)$-*strong growth property* asserts

$$D_f(x_{t+1}, x_t) \le \gamma_t^2 M/2\langle \nabla f(x_t), x_t - v_t \rangle^r,$$

with $M > 0$ and $r \in [0, 1]$.

### E.2.1 PRIMAL-DUAL CONVERGENCE $r = 1$

We start with the most obvious case $r = 1$, then (35) becomes:

$$A_t D_f(x_{t+1}, x_t) - a_t\langle \nabla f(x_t), x_t - v_t \rangle \le 2M\frac{a_t^2}{A_t}\langle \nabla f(x_t), x_t - v_t \rangle - a_t\langle \nabla f(x_t), x_t - v_t \rangle$$
$$= \langle \nabla f(x_t), x_t - v_t \rangle a_t(2M\frac{a_t}{A_t} - 1),$$

and hence choosing $a_t/A_t = \frac{1}{2M}$, which implies $\gamma_t = \frac{1}{M}$, the right-hand side becomes 0 and we obtain linear convergence with a rate $1/A_{t-1}(1 - \frac{1}{2M}) = 1/A_t$.

### E.2.2 PRIMAL-DUAL CONVERGENCE $r \in [0, 1)$

The situation is much more complicated for $r \in (0, 1)$. In this case (35) becomes:

$$A_t D_f(x_{t+1}, x_t) - a_t\langle \nabla f(x_t), x_t - v_t \rangle \le 2M\frac{a_t^2}{A_t}\langle \nabla f(x_t), x_t - v_t \rangle^r - a_t\langle \nabla f(x_t), x_t - v_t \rangle$$

We now apply Young's inequality with $p = \frac{1}{r}$, $q = \frac{1}{1-r}$, rebalancing with $\frac{a_t^r}{r^r}$, so that we obtain:

$$2M\frac{a_t^2}{A_t}\langle\nabla f(x_t), x_t - v_t\rangle^r - a_t\langle\nabla f(x_t), x_t - v_t\rangle$$

$$= 2M\frac{a_t^2}{A_t}\frac{r^r}{a_t^r}\frac{a_t^r}{r^r}\langle\nabla f(x_t), x_t - v_t\rangle^r - a_t\langle\nabla f(x_t), x_t - v_t\rangle$$

$$\leq \frac{\left(2M\frac{a_t^2}{A_t}\frac{r^r}{a_t^r}\right)^q}{q} + \frac{\left(\frac{a_t^r}{r^r}\langle\nabla f(x_t), x_t - v_t\rangle^r\right)^{1/r}}{1/r} - a_t\langle\nabla f(x_t), x_t - v_t\rangle$$

$$= (1-r)\left(2M\frac{a_t^2}{A_t}\frac{r^r}{a_t^r}\right)^{\frac{1}{1-r}} = (1-r)(2Mr^r)^{\frac{1}{1-r}}\left(\frac{a_t^{2-r}}{A_t}\right)^{\frac{1}{1-r}}$$

Summing the errors as customary, we obtain:

$$\frac{1}{A_T}\sum_{\ell=0}^{T}E_\ell \leq (1-r)(2Mr^r)^{\frac{1}{1-r}}\frac{1}{A_T}\sum_{t=0}^{T}\left(\frac{a_t^{2-r}}{A_t}\right)^{\frac{1}{1-r}},$$

which now needs to be estimated to derive the convergence rate.

**Estimation in the order** We will first do an approximate estimation, to recover the order of convergence and provide intuition. Suppose that $a_t = t^c$ and hence $A_t \approx \frac{t^{c+1}}{c+1} = \Theta(t^{c+1})$. With this we obtain:

$$(1-r)(2Mr^r)^{\frac{1}{1-r}}\frac{c+1}{T^{c+1}}\sum_{t=0}^{T}\left(\frac{t^{c(2-r)}}{t^{c+1}}\right)^{\frac{1}{1-r}}$$

$$= (1-r)(2Mr^r)^{\frac{1}{1-r}}\frac{c+1}{T^{c+1}}\sum_{t=0}^{T}\left(t^{c(2-r)-c-1}\right)^{\frac{1}{1-r}}$$

$$\approx (1-r)(2Mr^r)^{\frac{1}{1-r}}\frac{c+1}{T^{c+1}}\frac{T^{c-\frac{1}{1-r}+1}}{c-\frac{1}{1-r}+1} = (1-r)(2Mr^r)^{\frac{1}{1-r}}\frac{c+1}{c-\frac{1}{1-r}+1}T^{-\frac{1}{1-r}},$$

assuming $c > \frac{1}{1-r}$. Therefore we obtain for the total convergence rate:

$$G_t \leq \frac{A_0 G_0}{A_T} + (1-r)(2Mr^r)^{\frac{1}{1-r}}\frac{c+1}{c-\frac{1}{1-r}+1}T^{-\frac{1}{1-r}},$$

together with $1/A_T \approx \frac{c+1}{T^{c+1}} \leq \frac{c+1}{T^{\frac{1}{1-r}+1}}$, implies via a rather weak estimation

$$G_t \leq 2(1-r)(2Mr^r)^{\frac{1}{1-r}}\frac{c+1}{c-\frac{1}{1-r}+1}T^{-\frac{1}{1-r}},$$

**Specific choice of $a_t$** We will now refine the analysis from above via a specific choice of $a_t$ and $A_t$. We choose $a_t = a_0\binom{t+\ell-1}{t}$ and $A_t = a_0\binom{t+\ell}{t}$ similar to Lemma E.3. Note however the changed linear coupling $2a_t/A_t = \gamma_t$ that we use here:

$$(1-r)\left(2Mr^r\right)^{\frac{1}{1-r}}\frac{1}{A_T}\sum_{t=0}^{T}\left(\frac{a_t^{2-r}}{A_t}\right)^{\frac{1}{1-r}}$$

$$=(1-r)\left(2Mr^r\right)^{\frac{1}{1-r}}\frac{1}{A_T}\sum_{t=0}^{T}\left(\frac{a_t}{A_t}\right)^{\frac{1}{1-r}}a_t$$

$$=(1-r)\left(2Mr^r\right)^{\frac{1}{1-r}}a_0^{-1}\binom{T+\ell}{T}^{-1}\sum_{t=0}^{T}\left(\frac{\ell}{t+\ell}\right)^{\frac{1}{1-r}}a_0\binom{t+\ell-1}{t}$$

$$=(1-r)\left(2Mr^r\right)^{\frac{1}{1-r}}\binom{T+\ell}{T}^{-1}\sum_{t=0}^{T}\left(\frac{\ell}{t+\ell}\right)^{\frac{1}{1-r}}\binom{t+\ell-1}{t}$$

$$\leq(1-r)\left(2Mr^r\right)^{\frac{1}{1-r}}\binom{T+\ell}{T}^{-1}\frac{\ell^{\frac{1}{1-r}}}{(\ell-1)!}\sum_{t=0}^{T}t^{\ell-1-\frac{1}{1-r}}$$

$$=(1-r)\left(2Mr^r\right)^{\frac{1}{1-r}}\binom{T+\ell}{T}^{-1}\frac{\ell^{\frac{1}{1-r}}}{(\ell-1)!}\left(\frac{T^{\ell-\frac{1}{1-r}}}{\ell-\frac{1}{1-r}}+O\left(T^{\ell-1-\frac{1}{1-r}}\right)\right)$$

$$\leq(1-r)\left(2Mr^r\right)^{\frac{1}{1-r}}\frac{\ell^\ell}{T^\ell}\frac{\ell^{\frac{1}{1-r}}}{(\ell-1)!}\left(\frac{T^{\ell-\frac{1}{1-r}}}{\ell-\frac{1}{1-r}}+O\left(T^{\ell-1-\frac{1}{1-r}}\right)\right)$$

$$=(1-r)\left(2Mr^r\right)^{\frac{1}{1-r}}\frac{\ell^{\ell+\frac{1}{1-r}}}{(\ell-1)!}\left(\frac{T^{-\frac{1}{1-r}}}{\ell-\frac{1}{1-r}}+O\left(T^{-1-\frac{1}{1-r}}\right)\right)$$

$$=(1-r)\left(2Mr^r\right)^{\frac{1}{1-r}}\frac{\ell^{\ell+\frac{1}{1-r}}}{(\ell-\frac{1}{1-r})(\ell-1)!}T^{-\frac{1}{1-r}}+O\left(T^{-1-\frac{1}{1-r}}\right),$$

with $\ell>1+\frac{1}{1-r}$. Therefore we obtain for the total convergence rate:

$$G_t\leq\frac{A_0G_0}{A_T}+(1-r)\left(2Mr^r\right)^{\frac{1}{1-r}}\frac{\ell^{\ell+\frac{1}{1-r}}}{(\ell-\frac{1}{1-r})(\ell-1)!}T^{-\frac{1}{1-r}}+O\left(T^{-1-\frac{1}{1-r}}\right),$$

together with $1/A_T\leq\ell^\ell T^{-\ell}\leq\ell^\ell T^{-1-\frac{1}{1-r}}$, this implies

$$(1-r)\left(2Mr^r\right)^{\frac{1}{1-r}}\frac{\ell^{\ell+\frac{1}{1-r}}}{(\ell-\frac{1}{1-r})(\ell-1)!}T^{-\frac{1}{1-r}}+O\left(T^{-1-\frac{1}{1-r}}\right).$$

**Remark E.5** (Step-size choice and short steps). *Note, that we can always use short-steps instead of the $\frac{\ell}{t+\ell}$-step size by monotonicity and obtain the same rates.*

## F    EXPERIMENTS: APPENDIX

We performed additional experiments to test convergence and isolate the effect of the different components. Note that for the additional experiments below, we sometimes have used additional LMO calls for computing evaluation metrics for comparisons, so that the timings in the right columns are not as reliable and not comparable to the main experiments. Either way, the performance in iterations gives a clear picture here as the cost per iteration of all algorithms is quite comparable, apart from the adaptive variant which behaves more like line search but serves as a baseline only.

### PRIMAL-DUAL SHORT-STEPS

We conducted a series of experiments to evaluate the performance of the primal-dual short step method described in Section 4. Unfortunately, our findings indicate that this variant does not outperform the vanilla FW variant with short-steps (or line search). The primary reason for this effect appears to be the improved primal-dual gap measure, which results in a *smaller* step size, simply by how the step

size is computed. Consequently, this leads to slower convergence rates. This behavior is reminiscent of the lower bound discussed in (Guélat & Marcotte, 1986), and thus, it is not entirely unexpected. Moreover, by focusing on large primal-dual progress, the algorithm can overshoot in terms of primal progress, so that while the primal-dual progress is relatively large, we are still converging slowly (assuming that the primal-dual gap converges at a certain rate).

To illustrate this point, we present a typical experiment in Figure 3. This figure is representative of the results we observed across various tests, consistently showing that the primal-dual short step method converges at a similar rate to the vanilla FW variant. For completeness, we also include the results for the optimistic variant, which indeed outperforms the other variants.

### HEAVY BALL LOWER BOUND VS OPTIMISM

In our experiments, we also investigated whether the heavy ball lower bound is inherently stronger than the vanilla FW lower bound, which might have implied that optimism does not actually help at all. However, our findings indicate that this is not the case; see Figure 4 for a representative experiment. Specifically, when the heavy ball lower bound is applied to the vanilla FW variant, it actually performs worse than the standard FW gap in our tests. This observation strongly suggests that the superior performance of the optimistic variant is due to its trajectory rather than the strength of the lower bound itself.

### ADDITIONAL EXPERIMENTS

We present two additional experiments in Figures 5 and 6 that had to be relegated to the appendix due to space constraints. Also here the results are consistent with our findings in the main text.

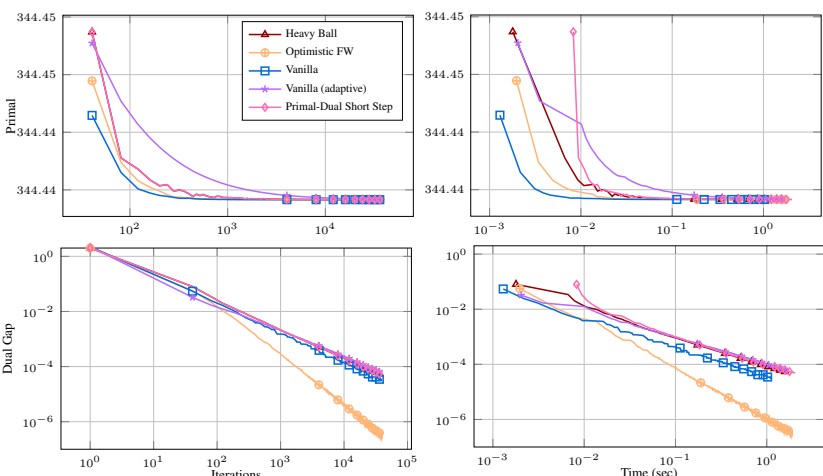

Figure 3: Comparison over the probability simplex of dimension $n = 1000$ with objective $f(x) = \|x - x_0\|_2^2$, where $x_0$ is a random point outside the probability simplex. We see that primal-dual short steps converge exactly as fast as the vanilla variant.

## G    OTHER EXPERIMENTAL SETTINGS

We also compared vanilla FW and Frank-Wolfe with adaptive line search of Pedregosa et al. (2020) (which it is currently the fastest non-active set variant in the FrankWolfe.jl package) with our optimistic variant on:

- 280 instances over standard vector LMOs from the FrankWolfe.jl package, which includes problem setups such as, e.g., sparse regression instances, approximate Caratheodory, etc.

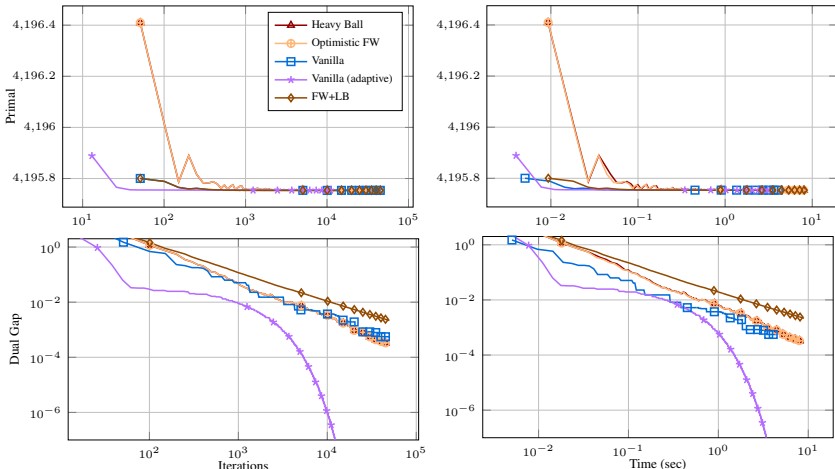

Figure 4: Comparison over the probability simplex of dimension $n = 1000$ with objective $f(x) = \|x - x_0\|_2^2$, where $x_0$ is a random point outside the probability simplex. FW+LB denotes the standard FW variant with the heavy ball lower bound. We can see that FW with the heavy ball lower bound converges no faster than the normal FW in dual gap.

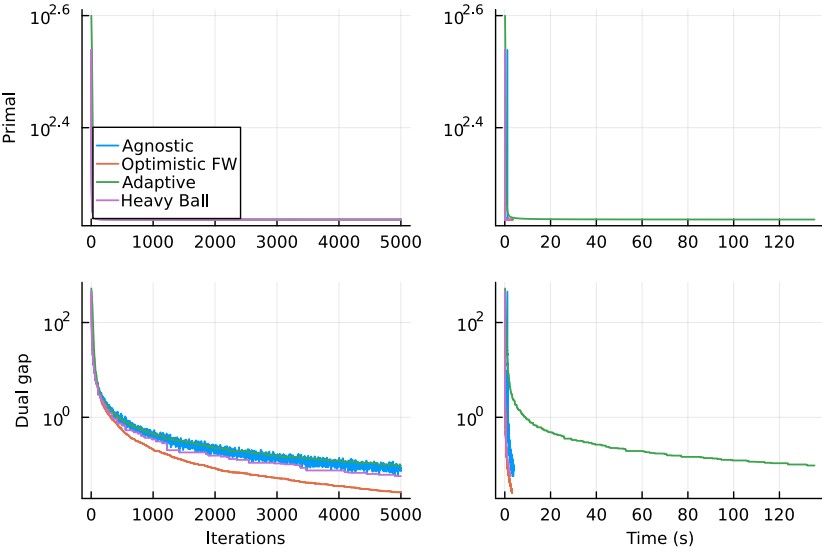

Figure 5: Comparison over the probability simplex and with a $1000 \times 1500$ matrix, with the negative log-likelihood of the portfolio returns. We can see that the optimistic variant converges faster than the other variants both in iterations and time.

- 70 instances over the Birkhoff polytope, capturing a convex version of the Quadratic Assignment Problem as well as general bad-conditioned quadratics that are challenging for Frank-Wolfe.

- 17 portfolio optimization instances from Carderera et al. (2021).

With a total of 367 computational experiments to further substantiate the performance of the optimistic FW algorithm. The experiments required roughly 61 compute hours, partially run in parallel on a M1 Ultra architecture running Julia 1.11.6.

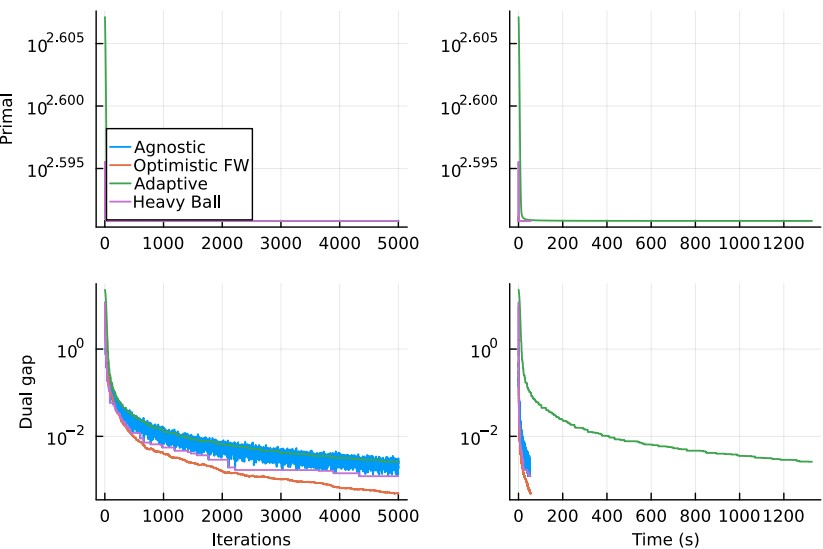

Figure 6: Comparison over the probability simplex and with a $1000 \times 1500$ matrix, with the negative log-likelihood of the portfolio returns. We can see that the optimistic variant converges faster than the other variants both in iterations and time.

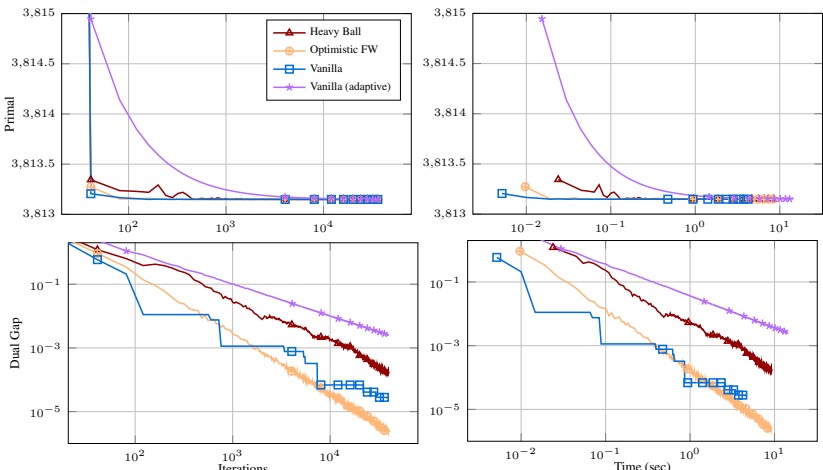

Figure 7: Comparison over the probability simplex of dimension $n = 100$ with objective $f(x) = \|Ax - b\|_2^2$, where $A$ and $b$ are random. We can see that the optimistic variant converges faster than the other variants both in iterations and time.

We now detail the experiments. As customary, we break out complexities by feasible regions as they typically drive the cost of Frank-Wolfe algorithms. We generated instances of varying sizes (10, 100, 500, 1000) and created 10 random instances for each size in the case of the vector LMOs. In the case of the Birkhoff polytope we used smaller sizes (5, 10, 15, 20, 30, 40, 50) as the matrices are quadratic in the size and the Hungarian method is significantly more expensive than the LMOs for the other feasible regions.

We imposed an iteration limit of 20,000 iterations and the following tables summarize the findings. As customary we report (shifted) geometric means both for time and iterations (that is, number of LMO calls). For more information about performance measurements for Frank-Wolfe algorithms, see also the Besançon, M. et al (2025). Examples and Tests are from the FrankWolfe.jl package.

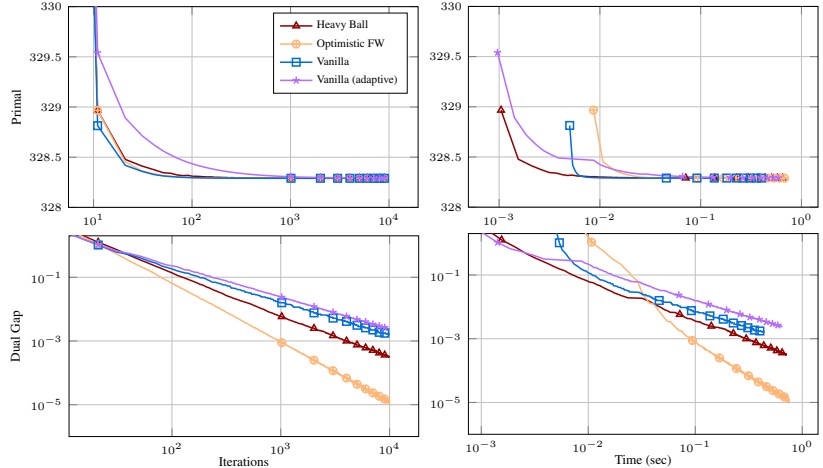

Figure 8: Comparison over $k$-sparse polytope of dimension $n = 1000$ and $k = 10$ with objective $f(x) = \|x - x_0\|_2^2$, where $x_0$ is a random point outside the $k$-sparse polytope. We can see that the optimistic variant converges faster than the other variants both in iterations and time.

Table 1: Shifted Geometric Means (shift parameter: 1.0).

| LMO (instances) | Agno. Iter | Agno. Time | OFW Iter | OFW Time | Adap. Iter | Adap. Time |
|---|---|---|---|---|---|---|
| k_sparse_10 (40) | 1675.08 | 6.02 | **716.42** | **4.50** | 2187.30 | 16.50 |
| k_sparse_20 (40) | 1186.59 | 4.16 | **492.23** | **2.87** | 1485.73 | 11.33 |
| k_sparse_30 (40) | 1306.53 | 2.39 | **518.05** | **1.55** | 1627.67 | 6.76 |
| l1_norm_ball (40) | 1438.42 | 2.13 | **982.91** | **2.10** | 1666.70 | 6.16 |
| l2_norm_ball (40) | 1754.52 | 0.06 | 440.65 | 0.04 | **72.94** | **3.93e-03** |
| prob_simplex (40) | 307.08 | 2.52 | 481.95 | 3.96 | **73.58** | **1.18** |
| unit_simplex (40) | 371.89 | 2.70 | 545.18 | 4.44 | **44.79** | **0.83** |
| birkh_sc10 (10) | 4937.21 | 0.11 | **2329.91** | **0.04** | 7956.67 | 0.32 |
| birkh_sc15 (10) | 6608.77 | 0.31 | **2841.41** | **0.12** | 1.09e+04 | 1.29 |
| birkh_sc20 (10) | 7742.95 | 0.69 | **3279.43** | **0.28** | 1.19e+04 | 3.07 |
| birkh_sc30 (10) | 9748.16 | 2.16 | **4034.34** | **0.93** | 1.49e+04 | 11.84 |
| birkh_sc40 (10) | 1.09e+04 | 4.89 | **4394.79** | **2.09** | 1.60e+04 | 30.27 |
| birkh_sc5 (10) | 1722.72 | 0.07 | **1103.18** | **0.01** | 2414.81 | 0.03 |
| birkh_sc50 (10) | 1.18e+04 | 9.36 | **4769.02** | **4.02** | 1.72e+04 | 63.78 |

Table 2: Portfolio Optimization Instances. Shifted Geometric Means (shift parameter: 1.0).

| Size (instances) | Agno. Iter | Agno. Time | OFW Iter | OFW Time | Adap. Iter | Adap. Time |
|---|---|---|---|---|---|---|
| 1000×1200 (4) | **210.58** | 0.21 | 280.65 | **0.09** | 832.59 | 14.33 |
| 1000×1500 (5) | **54.47** | 0.20 | 60.56 | **0.18** | 138.17 | 3.60 |
| 1000×800 (4) | **255.71** | **0.05** | 300.85 | 0.06 | 860.12 | 9.91 |
| 1500×1500 (1) | 1817.00 | 0.95 | **1151.00** | **0.64** | 4628.00 | 131.53 |
| 5000×2000 (2) | 299.59 | 2.94 | **259.29** | **2.36** | 597.46 | 68.76 |
| 5000×5000 (1) | 2102.00 | 21.40 | **1390.00** | **14.73** | 4982.00 | 1417.25 |