# OpenReview forum: "Beyond Short Steps in Frank-Wolfe Algorithms"
_ICLR.cc/2026/Conference — ICLR 2026 Poster_

### Official Review · Reviewer_C2jy · 2025-10-30

**Soundness:** 4
**Presentation:** 3
**Contribution:** 3
**Rating:** 8
**Confidence:** 3

**Summary:**

This paper revisits the Frank–Wolfe (FW) family of algorithms and develops two main innovations: (i) optimistic Frank–Wolfe (OFW): A new FW variant that incorporates optimism from online learning, predicting future gradients to adaptively refine updates. The algorithm enjoys a primal–dual convergence analysis and retains the rate under convex smooth settings. (ii) primal–dual short steps: A generalization of the classical “short-step” rule that maximizes primal–dual progress instead of just primal progress. This approach yields an interpretable stopping criterion based on a computable primal–dual gap and extends naturally to gradient descent.

The paper provides rigorous analyses, unifying online learning and Frank–Wolfe perspectives via the optimistic mirror descent/FTRL framework, and demonstrates the practical advantages of OFW through experiments on convex quadratic problems and polytopal constraints.

**Strengths:**

1. The incorporation of optimism into Frank–Wolfe algorithms is both original and theoretically well-motivated. It bridges FW with the online-to-batch conversion framework (Cutkosky, 2019), enriching the algorithmic toolbox for projection-free methods.

2. The proposed analysis elegantly derives both convergence guarantees and computable stopping criteria from a primal–dual perspective. The connection between FW, heavy-ball FW, and gradient descent is clarified through shared gap-based arguments.

3. The theoretical results are sound and internally consistent. The paper’s exposition (especially Section 3-4) carefully motivates each step-size rule and provides transparent algorithmic derivations.

4. Experiments on probability simplices and $k$-sparse polytopes convincingly show that the optimistic variant converges faster (both in iteration and wall-clock time) than standard and heavy-ball FW variants. The results are reproducible and coded in Julia.

**Weaknesses:**

1. The experiments are small-scale, focusing on synthetic convex problems. More realistic applications (e.g., structured prediction, optimal transport, or matrix completion) would better support the method’s practical relevance.

2. Despite being theoretically elegant, the primal–dual short-step variant performs comparably or slightly worse than vanilla FW with line search, suggesting limited practical value.

3. The optimistic connection could be explained more clearly and the connection to predictive gradient methods could be emphasized further.

4. The paper is dense with notations, which can make it difficult for non-specialists to follow. A clearer summary table or pseudocode for each variant would help.

**Questions:**

1. Can the optimism mechanism extend to nonconvex or stochastic FW settings?
2. How does the optimistic FW relate to momentum-based FW (e.g., Heavy-Ball FW) beyond shared lower bounds?
3. Could the primal–dual gap be efficiently estimated without full LMO recomputation at each iteration?
4. Do you foresee any extension to affine-invariant step-size strategies as in Pena (2023)?

**Details Of Ethics Concerns:**

N/A.

---

> ### Author Response · Authors · 2025-11-20
> **Rebuttal**
>
> We thank the reviewer for their time and their thoughtful review. We appreciate the highlights of the strengths of our work. Regarding the other comments:
>
> + Experiments
>     + Please see the global comment to other reviewers including the results of many more experiments showing the consistency of the empirical advantages of our algorithm. We also performed an experiment with real data.
>
> + Presentation: Explanations on optimism and highlighting connections
>     + we will do our best to incorporate some more comments to better explain the connection with optimism and predictive gradient methods. We will help the reader with a notation table or some other mechanism.
>
> ### Questions
>
> + Can the optimism mechanism extend to nonconvex or stochastic FW settings?
>    + Since the deadline, we generalized our analysis for the stochastic case. We are not sure about the nonconvex case but it is definitely something we are curious about.
>
> + Do you foresee any extension to affine-invariant step-size strategies as in Pena (2023)?
>     + This is an interesting suggestion. We have to look into this carefully to see whether the approach can be generalized. In principle though this should be possible especially under Pena's growth settings; in fact some of the analyses in the appendix have been already inspired by this point of view.
>
> + How does the optimistic FW relate to momentum-based FW (e.g., Heavy-Ball FW) beyond shared lower bounds?
>     + They are very different in spirit as far as we can tell. Heavy ball does not exploit the fact that if two consecutive gradients are similar, then the convergence rate improves to the best of our knowledge. This is also consistent with the findings from our computational experiments.
>
> + Could the primal–dual gap be efficiently estimated without full LMO recomputation at each iteration?
>     + Yes, in Remark B.6 we provide a simple bound on the primal-dual gap that comes for free for our algorithm. It consists of bounding away the only term depending on the minimizer.

---

> > ### Comment · Reviewer_C2jy · 2025-11-27
> > **Thank you**
> >
> > I would like to thank the authors for the response. All my questions are addressed. I will keep my positive score.

---

### Official Review · Reviewer_CvWP · 2025-11-01

**Soundness:** 3
**Presentation:** 2
**Contribution:** 3
**Rating:** 6
**Confidence:** 4

**Summary:**

This paper introduces new techniques for improving Frank-Wolfe (FW) algorithms by incorporating function smoothness beyond traditional short steps. The authors propose two main innovations. First it proposed the Optimistic Frank-Wolfe Algorithm (OFW), a novel method inspired by optimistic online learning, leveraging predicted gradients to improve convergence in smooth convex optimization. It adapts to changing curvature and achieves a provable 1/t^2 like primal-dual convergence rate. Second it introduced the Primal-Dual Short Steps (PDSS), a new class of step-size rules that optimize the primal-dual gap directly, leading to theoretically justified and generalizable improvements. The same principle extends to gradient descent. Experiments over standard convex benchmarks (simplex, k-sparse polytopes, portfolio optimization) show that the optimistic variant consistently outperforms classical and heavy-ball FW methods, both in iteration count and runtime.

**Strengths:**

This submission has conceptual innovation. The introduction of optimism into the FW setting is novel and well-motivated through online learning theory (OMD/FTRL frameworks). The primal-dual short-step idea elegantly unifies step-size selection with primal-dual analysis, offering tighter convergence and computable stopping criteria. The authors presented the rigorous theoretical framework. The analysis is carefully constructed through primal-dual gap bounds, clearly improving on classical FW and heavy-ball variants. Results from Theorems and Propositions provide clean, interpretable convergence bounds with constant factors comparable to or better than prior work.

The authors also conduct the numerical experiments using standard benchmarks implemented in Frank Wolfe, ensuring reproducibility. The optimistic algorithm shows steeper log–log slopes (faster empirical order of convergence) and strong wall-clock performance improvements. The empirical results from the numerical experiments are consistent with the theoretical analysis.

**Weaknesses:**

There are no major weakness for this submission.

On minor weakness if that this submission has limited experimental scopes. Only convex smooth problems are tested; no constrained stochastic or non-convex scenarios. The proposed primal-dual short steps did not outperform standard heuristics in practice — acknowledged but not deeply analyzed. It is suggested to add more numerical experiments with more experimental scopes like including non-smooth and stochastic benchmarks to demonstrate generality and comparing against projection-free adaptive gradient methods or away-step FW variants to position contributions more competitively.

**Questions:**

There are no other questions.

---

> ### Author Response · Authors · 2025-11-20
> **Rebuttal**
>
> We thank the reviewer for their time and we appreciate that our conceptual innovation and results were acknowledged.
>
> Regarding the concerns, we performed several new experiments, please see the global comment we made to all reviewers for details. Our new experiments showcase the consistency of the empirical advantages of our algorithm across various tests. Finally, we note that we had included portfolio optimization experiments in our submission, that are not smooth.
>
> If you are happy with the provided answer we would really appreciate if you would adjust your score accordingly.

---

> > ### Comment · Reviewer_CvWP · 2025-11-27
> >
> > Thanks for your rebuttal. I remain my original score.

---

### Official Review · Reviewer_62CQ · 2025-11-01

**Soundness:** 3
**Presentation:** 3
**Contribution:** 3
**Rating:** 6
**Confidence:** 3

**Summary:**

The paper proposes an optimistic Frank-Wolfe algorithm that leverages optimistic online learning methods (FTRL, OMD). In particular, one can substitute the LMO oracle part into optimistic FTRL/OMD on linear losses to construct the proposed OFW. The OFW analysis establishes a classical $O(1/t)$ rate of convergence for the primal-dual gap via online-to-batch conversion and shows better performance in empirical studies, especially in terms of the dual gap. The paper also suggests a primal-dual short step schedule designed from a descent-lemma-like inequality tailored for the Frank-Wolfe (primal-dual) gap. This better suits FW type algorithms than classical step sizes that come from ‘primal’ descent-lemmas (i.e., smoothness) and provides a stopping criterion computable on the fly.

**Strengths:**

- The writing is clear and easy to read overall. The ideas of each sections are well presented through selected key inequalities from the proof, which makes a much better reading experience.
- The paper includes numerical experiments that support the theoretical results.

**Weaknesses:**

See **Questions.**

**Questions:**

- The second term in the proposed upper bound for optimistic FW (in Theorem 3.1) is $\frac{4LD^2}{t+1}$ which looks identical to that of vanilla FW, but in the experiments optimistic FW clearly looks faster. Could there be a way to theoretically show a strict improvement over FW/HB-FW in terms of either faster convergence under certain settings (sparse problems, etc) or guaranteed convergence in a broader setting with weaker assumptions or maybe something else? (In simple online learning settings, FTRL and OMD can have constant improvements in the regret upper bound compared to vanilla online GD in certain cases, e.g., the function $f$ has sparse gradients, if I am correct.)
- Is it a bad idea to choose $L_2$ or $L_1$ (or maybe combined) norm regularizers for optimistic FW, instead of the polytope indicator? I think this might sometimes better capture the benefits of using FTRL/OMD type updates, while using an indicator as the regularizer seems to make less distinction (though not exactly equivalent) with vanilla FW. What can you expect if we use different regularizers, or is there a reason why the paper does not consider these?
- What exactly are the benefits we get from using primal-dual step sizes? What I can see is that we can drop the constant $4$ from the upper bound of $G_t$ if we use GD with primal-dual step sizes (or line search), but is this the best we can get from this? Also, is there a way to find a ‘primal-dual-optimized’ $a_t$ for the optimistic FW versions? While merging the two ideas might be irrelevant or unbeneficial, I still think there should have been bits of explanation about why this is the case, and if these are two completely separate ideas, it seems unclear how primal-dual step sizes can be useful.

---

> ### Author Response · Authors · 2025-11-20
> **rebuttal**
>
> We thank the reviewer for their time. Below, we reply to the questions raised.
>
> + Could there be a way to theoretically show a strict improvement over FW/HB-FW in terms of either faster convergence or weaker assumptions? (FTRL and OMD can have improvements in some cases)
>     + This is a great question and a promising direction of future research. We had started taken steps in this direction but we have not come up with a solution for this interesting question yet.
>
>
> + Explanation about optimism and primal-dual step sizes
>     + We provide ideas to go beyond short steps in Frank-Wolfe algorithms and are investigating whether an optimized primal-dual step can be obtained for the optimistic algorithm.
>
> + Using regularizers other than the set indicator?
>     + One of the main advantages of FW algorithms consists of being projection free, obtaining an advantage over projected GD methods (likewise OMD/FTRL) when (Bregman) projections are very expensive in comparison to linear optimization over the feasible set. This is the main reason of our choice of the regularizer. If the Bregman projection associated to other regularizers is cheap, one may consider using it.
>
> +  Benefits from primal-dual step sizes. Is this the best we can get from them?
>     + The primal-dual step sizes also aim to decrease the primal-dual gap as fast as possible. We are investigating whether there are more benefits of this idea, but we hope that in future work we can provide a statement about these step sizes being better for decreasing a computable stopping criterion that is the primal-dual gap, as opposed to trying to just optimize the gap since that is not computable in general along the optimization trajectory.
>
> Please also take a look at the new set of experiments showcasing the consistency of the empirical advantages of our algorithm across various tests.

---

### Official Review · Reviewer_n1pE · 2025-11-03

**Soundness:** 3
**Presentation:** 2
**Contribution:** 2
**Rating:** 4
**Confidence:** 3

**Summary:**

This paper studies an optimistic variant of the Frank-Wolfe method and establishes a worst-case convergence guarantee in terms of the primal dual gap $G_t = f(x_t) - L_t$, where $L_t$ is a computable lower bound on the optimal value $f(x_*)$. The analysis focuses on the primal-dual gap because it can be evaluated and used as a stopping criterion, unlike the primal gap $f(x_t) - f(x_*)$. In particular, the authors propose Algorithm 2, which uses the short step size in (5) to ensure a monotonic decrease of the primal-dual gap. The paper further demonstrates the generality of the approach by showing that it also applies to gradient descent.

**Strengths:**

Regardless of the weaknesses mentioned below, I believe this paper, by proposing a new optimistic variant of the Frank-Wolfe method with a convergence guarantee on the computable measure, has its own merit, and warrants further investigation in this direction.

**Weaknesses:**

- I agree that the primal-dual gap can serve as a practical stopping criterion, but I don't follow the authors' claim that this justifies the need for a method with a guaranteed convergence rate for that gap. First, the primal-dual gap is not a tight bound on the primal gap $f(x_t) - f(x_*)$, and it can be computed regardless of whether we have a theoretical guarantee on its decrease. Moreover, the paper does not show that existing methods do not efficiently decrease the primal-dual gap (in theory).

- I also find the repeated claim that the proposed optimistic approach better adapts to the environment unconvincing. I would appreciate a more precise and unambiguous explanation of what type of adaptivity is meant here.

- For the above reasons, I am not fully convinced by the motivation or the importance of the contribution.

**Questions:**

- Line 43: What do you mean by "corresponding convergence rate need to be heuristically estimated"?
- Line 46: What do you mean by "enhancing the convergence properties"? I don't see any results that supports this claim.
- Line 65: Could you justify how your method adapt effectively to varying conditions and provides a robust analysis?
- Line 78: As written, it seems to suggest that your method improves the theoretical constant, although you refer Appendix E. I recommend explicitly stating that this improvement is observed empirically.
- Lines 230-234 with line 7 in Algorithm 2: This part is particularly difficult to follow.
- Figure 1: Why does the optimistic method much more efficient per iteration compared to the plain one, given that Algorithm 2 is more complex? I also recommend plotting the primal gap $f(x_t) - f(x_*)$, rather than $f(x_t)$.

---

> ### Author Response · Authors · 2025-11-20
> **Rebuttal**
>
> We thank the reviewer for their time and we deeply appreciate the kind words regarding the merits of our paper and original idea as well as the potential of the research direction that our ideas open up.
>
> We reply to your comments below:
>
> + I would appreciate a more precise and unambiguous explanation of what type of adaptivity is meant here for optimism. (also, question about line 46 and line 65)
>     + We mean two very precise things, which we will highlight in a comment in the paper. (A) optimism allows for a speed up in the convergence rates whenever the hint is good. We use the previous gradient to predict the next gradient and whenever these predictions are good enough, the method improves. The rate provided in the theorem is worse-case only, but in the second line of Equation (21) one can see how the terms appearing in the convergence rates becomes smaller when the prediction is good, which is a phenomenon that previous FW algorithms do not exploit. (B) experiments show consistent better performance in different scenarios (we strongly believe this is due to reason (A)).
>
>
> + Line 43: What do you mean by "corresponding convergence rate need to be heuristically estimated"?
>     + Previous approaches in the literature to show convergence rates, work by showing the rates via induction by first proposing a desired rate and then showing it, leaving it open how this guess was obtained in the first place and not allowing to transfer this knowledge to other analysis. In contrast, in our analysis the best step size and the corresponding convergence rate are obtained naturally from the analysis, i.e., they are a natural choice rather than a nondeterministic guess. This is a general technique that can be used in future works and not only speeds up finding the "correct" optimal proofs, but can also be used as a tool to discover rates in previously unknown scenarios.
>
> + Line 78
>     + We will emphasize and explicitly state that the improvement is observed empirically, as per the reviewer's suggestion.
>
> + Lines 230-234 with line 7 in Algorithm 2: This part is particularly difficult to follow.
>     + We will write it more clearly, we are referring to Line 7 of Algorithm 2, which we will reference. It means that our proof with rates of convergence still goes through if we implement any heuristic as part of line 7.
>
> + Why is the optimistic method much more efficient per iteration compared to the plain one, given that Algorithm 2 is more complex?
>     + While the algorithm may appear more complex, in terms of actual computational complexity each iteration is about the same as the one for the plain FW algorithm; we only "average" along the way. At the same time, we observe more progress is made per-iteration. The key is that our method exploits the fact that if we can approximately predict the next gradient, then convergence improves: basically we do not pay a baseline error but the error between the prediction and the true gradient. More precisely, we use the previous gradient as the approximate prediction for the next one, so if gradients are slowly varying, we get an improvement (second line of equation 21), which is not exploited by the plain FW algorithm.
>
> + I also recommend plotting the primal gap. f(x_t)-f(x*) instead of f(x_t)
>     + Good point, we will consider this for the revision of the graphs.
>
> Please let us know if there are still open questions, which we will happily answer. Otherwise we would appreciate if you could adjust the score accordingly.

---

### Author Response · Authors · 2025-11-20
**New experiments**

We make this global comment to all reviewers since a few asked about a more broad set of experiments We performed additional computational experiments with an early implementation of the optimistic FW algorithm compatible with the FrankWolfe.jl package. Our timings are going to improve even further when some debug code is removed. Currently for example the optimistic gap function is computed from scratch in every iteration for numerical accuracy. This will be replaced by a recursive update in the final version. Nonetheless, even now the optimistic FW variant is very competitive and outperforms Frank-Wolfe with adaptive line search in most cases.

We compared vanilla FW (of Frank and Wolfe, however using the tuned implementation in FrankWolfe.jl) and Frank-Wolfe with adaptive line search of of Pedregosa et al. (2020) (currently fastest non-active set variant in the FrankWolfe.jl package) with the optimistic variant on:

+ 280 instances over standard (vector) LMOs from the FrankWolfe.jl package (this includes problem setups such as, e.g., sparse regression instances, approximate caratheodory, etc)
+ 70 instances over the Birkhoff polytope (capturing a convex version of the Quadratic Assignment Problem as well as general bad-conditioned quadratics that are challenging for Frank-Wolfe)
+ 17 portfolio optimization instances from "Carderera, A et al. (2021). Simple steps are all you need: Frank-Wolfe and generalized self-concordant functions. Proceedings of NeurIPS." that the authors genereously provided us with.

With a total of 367 computational experiments to further substantiate the performance of the optimistic FW algorithm. The experiments required roughly 61 compute hours, partially run in parallel on a M1 Ultra architecture running Julia 1.11.6.

We detail the experiments below; as customary we break out complexities by feasible regions as they typically drive the cost of Frank-Wolfe algorithms.
Experimental Design

We generated instances of varying sizes (10, 100, 500, 1000) and created 10 random instances for each size in the case of the vector LMOs. In the case of the Birkhoff polytope we used smaller sizes (5, 10, 15, 20, 30, 40, 50) as the matrices are quadratic in the size and the Hungarian method is significantly more expensive than the LMOs for the other feasible regions.

We imposed an iteration limit of 20,000 iterations and the following tables summarize the findings. As customary we report (shifted) geometric means both for time and iterations (= number of LMO calls); we also computed the performance profiles of these runs, however this time around it seems cannot share images in the rebuttal. For more information about performance measurements for Frank-Wolfe algorithms, see also the Besançon, M. et al (2025).
Examples and Tests from the FrankWolfe.jl package

---

> ### Author Response · Authors · 2025-11-20
> **New experiments (continuation)**
>
> + Shifted Geometric Means (shift parameter: 1.0)
>
> | LMO (instances)        | Agnostic Iter | Agnostic Time | Optimistic FW Iter | Optimistic FW Time | Adaptive Iter | Adaptive Time |
> |------------------------|---------------|----------------|--------------------|---------------------|---------------|----------------|
> | k_sparse_10 (40)       | 1675.08       | 6.02           | **716.42**         | **4.50**            | 2187.30       | 16.50         |
> | k_sparse_20 (40)       | 1186.59       | 4.16           | **492.23**         | **2.87**            | 1485.73       | 11.33         |
> | k_sparse_30 (40)       | 1306.53       | 2.39           | **518.05**         | **1.55**            | 1627.67       | 6.76          |
> | l1_norm_ball (40)      | 1438.42       | 2.13           | **982.91**         | **2.10**            | 1666.70       | 6.16          |
> | l2_norm_ball (40)      | 1754.52       | 0.06           | 440.65             | 0.04                | **72.94**     | **3.93e-03**  |
> | prob_simplex (40)      | 307.08        | 2.52           | 481.95             | 3.96                | **73.58**     | **1.18**      |
> | unit_simplex (40)      | 371.89        | 2.70           | 545.18             | 4.44                | **44.79**     | **0.83**      |
> | birkhoff_scale10 (10)  | 4937.21       | 0.11           | **2329.91**        | **0.04**            | 7956.67       | 0.32          |
> | birkhoff_scale15 (10)  | 6608.77       | 0.31           | **2841.41**        | **0.12**            | 1.09e+04      | 1.29          |
> | birkhoff_scale20 (10)  | 7742.95       | 0.69           | **3279.43**        | **0.28**            | 1.19e+04      | 3.07          |
> | birkhoff_scale30 (10)  | 9748.16       | 2.16           | **4034.34**        | **0.93**            | 1.49e+04      | 11.84         |
> | birkhoff_scale40 (10)  | 1.09e+04      | 4.89           | **4394.79**        | **2.09**            | 1.60e+04      | 30.27         |
> | birkhoff_scale5 (10)   | 1722.72       | 0.07           | **1103.18**        | **0.01**            | 2414.81       | 0.03          |
> | birkhoff_scale50 (10)  | 1.18e+04      | 9.36           | **4769.02**        | **4.02**            | 1.72e+04      | 63.78         |
>
>
> **Portfolio Optimization Instances**
>
> + Shifted Geometric Means (shift parameter: 1.0)
>
> | Size (instances) | Agnostic Iter | Agnostic Time | Optimistic FW Iter | Optimistic FW Time | Adaptive Iter | Adaptive Time |
> |------------------|---------------|----------------|--------------------|---------------------|---------------|----------------|
> | 1000x1200 (4)    | **210.58**    | 0.21           | 280.65             | **0.09**            | 832.59        | 14.33         |
> | 1000x1500 (5)    | **54.47**     | 0.20           | 60.56              | **0.18**            | 138.17        | 3.60          |
> | 1000x800 (4)     | **255.71**    | **0.05**       | 300.85             | 0.06                | 860.12        | 9.91          |
> | 1500x1500 (1)    | 1817.00       | 0.95           | **1151.00**        | **0.64**            | 4628.00       | 131.53        |
> | 5000x2000 (2)    | 299.59        | 2.94           | **259.29**         | **2.36**            | 597.46        | 68.76         |
> | 5000x5000 (1)    | 2102.00       | 21.40          | **1390.00**        | **14.73**           | 4982.00       | 1417.25       |

---

### Meta-Review · Area_Chair_CR6e · 2026-01-07

**Summary:**

This paper proposes a novel Frank-Wolfe method with the concept of optimism (Rakhlin & Sridharan, 2013; Steinhardt & Liang, 2014). The authors show convergence guarantees for their method. The reviewers appreciate the contributions of this paper, in terms of theoretical results and the presentations.

**Reviewer Concerns:**

Some concerns are:
- The claim that the proposed optimistic approach better adapts to the environment is not really convincing. While the authors provided an answer with intuition, I think the claim is still somewhat ambiguous and not theoretically justified. Also, it seems that the authors missed the question regarding the primal-dual gap.
- About limited experiments, which the authors responded with additional experiments.

**Reviewer Scores:**

The reviewer n1pE may or may not increase the score as their concerns are addressed partially.

---

### Decision · Program_Chairs · 2026-01-26

Accept (Poster)